# Semantic Representations of Mathematical Expressions in a Continuous Vector Space

**Neeraj Gangwar**  *gangwar2@illinois.edu*
*Electrical and Computer Engineering*
*University of Illinois Urbana-Champaign*

**Nickvash Kani**  *kani@illinois.edu*
*Electrical and Computer Engineering*
*University of Illinois Urbana-Champaign*

**Reviewed on OpenReview:** *https://openreview.net/forum?id=EWPA9TZcUy*

## Abstract

Mathematical notation makes up a large portion of STEM literature, yet finding semantic representations for formulae remains a challenging problem. Because mathematical notation is precise, and its meaning changes significantly with small character shifts, the methods that work for natural text do not necessarily work well for mathematical expressions. This work describes an approach for representing mathematical expressions in a continuous vector space. We use the encoder of a sequence-to-sequence architecture, trained on visually different but mathematically equivalent expressions, to generate vector representations (or embeddings). We compare this approach with a structural approach that considers visual layout to embed an expression and show that our proposed approach is better at capturing mathematical semantics. Finally, to expedite future research, we publish a corpus of equivalent transcendental and algebraic expression pairs.

## 1 Introduction

Despite there being well-established search technologies for most other modes of data, the processing of mathematical content remains an open problem (Larson et al., 2013). Effective search technologies require semantically rich and computationally efficient representations of mathematical data. Most equation embedding methods have focused on establishing a homomorphism between an equation and its surrounding mathematical text (Zanibbi et al., 2016; Krstovski & Blei, 2018). While this approach can help find equations used in similar contexts, it is less effective in the following situations: (1) surrounding text may be limited, for example, consider math textbooks that contain equations with minimal explanation, and (2) in scientific literature, an equation may be used in a variety of disciplines with different contexts, and encoding equations based on textual context may hamper cross-disciplinary retrieval.

We argue that context *alone* is not sufficient for finding representations of mathematical content, and embedding methods must understand equations in addition to the surrounding context. To this end, we present a novel embedding method to generate semantically rich representations of mathematical formulae without the aid of natural language context. In our proposed approach, *we train a sequence-to-sequence model on equivalent expression pairs* and use the trained encoder to generate vector representations or embeddings. Figure 1 shows an example of our approach that embeds expressions according to their semantics. We compare our proposed approach with a structural encoder, that considers the layout structure of an expression, on a variety of tasks to determine the usefulness of each. Furthermore, we compare our semantic embedding method with two prior works proposed for an analogous problem of clustering equivalent expressions: EqNet (Allamanis et al., 2017) and EqNet-L (Liu, 2022), further proving our model's ability to capture semantics.

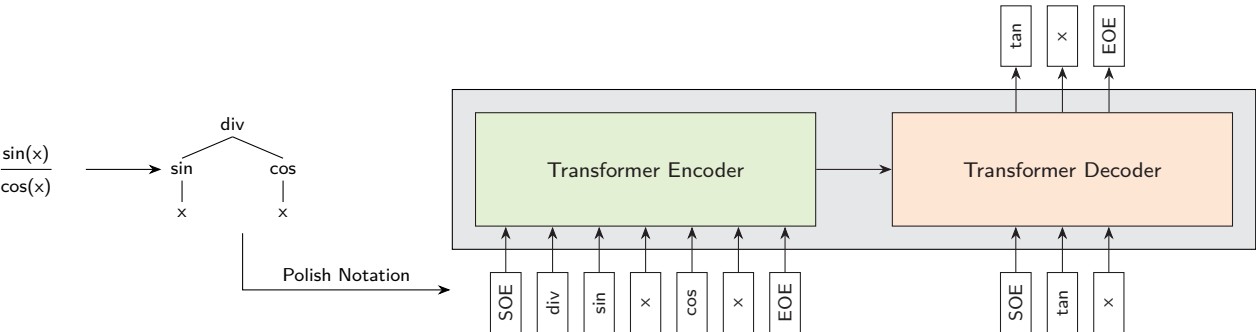

Figure 1: An overview of our approach. The model is trained to generate an expression mathematically equivalent to the input. For example, given an input expression $\frac{\sin(x)}{\cos(x)}$, the model learns to generate $\tan(x)$. We use max pooling on the hidden states of the last encoder layer corresponding to the input tokens and use the result as the *continuous vector representation* of the input expression. Here, "SOE", "EOE", and "div" represent the start token, the end token, and the division operation, respectively.

The contributions of our work are threefold:

1. We show that a sequence-to-sequence (SEQ2SEQ) model can learn to generate expressions that are mathematically equivalent to the input.

2. We use the encoder of such a model to generate semantically rich vector representations of mathematical expressions. They are better at clustering and retrieving similar mathematical expressions. We expand on the notion of semantics subsequently.

3. We publish a corpus of equivalent transcendental and algebraic expression pairs that can be used to develop more complex mathematical embedding approaches.

In this work, we consider the SEMVEC datasets that have previously been used by EQNET and EQNET-L to infer if representations are semantically rich. In addition, we perform distance analysis and embedding algebra and analyze embedding plots. For distance analysis, the similarity between two expressions is defined using the tree edit distance (Zhang & Shasha, 1989) between their operator trees. We ignore constant additions and multiplications while computing the tree edit distance and put emphasis on the operators. This is achieved by considering $f(x)$ and $af(x) + b$ to be at a distance of zero when $a$ and $b$ are constants.

We end this manuscript with a comprehensive study of our proposed approach, detailing its potential for general information processing and retrieval tasks and noting its limitations. The datasets and source code are available on GitHub. [1]

## 2 Related Work

While information processing for mathematical expressions is still a relatively new field, other groups have attempted to create embedding methods for mathematical content.

Starting with embedding schemes for mathematical *tokens*, Gao et al. (2017) embedded mathematical symbols using WORD2VEC (Mikolov et al., 2013) on the Wikipedia corpus. They created a mathematical token embedding scheme (SYMBOL2VEC) in which tokens used in similar contexts (sin & cos, = & ≈) were grouped together. Using these symbol embeddings, they extended their approach to embed entire formulae using the paragraph-to-vector approach (FORMULA2VEC). More commonly, other approaches have attempted to extract textual descriptors of equations using surrounding text and use pre-trained embeddings of those textual keywords to create an embedding of the equation in question (Kristianto et al., 2014; Schubotz et al., 2017).

---

[1] https://github.com/mlpgroup/expemb

Expanding token descriptors to represent equations is more complex. Schubotz et al. (2016) created equation descriptors by combining the textual keyword descriptors of mathematical tokens included within an expression. By organizing the equation descriptors into keyword pairs, clustering algorithms grouped equations according to Wikipedia categories. In a similar approach, Kristianto et al. (2017) viewed equations as dependency relationships between mathematical tokens. Nominal natural language processing (NLP) methods were used to extract textual descriptors for these tokens, and the equation was transformed into a dependency graph. A combination of interdependent textual descriptors allowed the authors to derive better formula descriptors that were used by indexers for retrieval tasks.

Alternatively, multiple groups have attempted to represent equations as feature sets; sequences of symbols that partially describe an equation's visual layout (Zanibbi et al., 2015; Fraser et al., 2018). Krstovski & Blei (2018) used WORD2VEC in two different manners to find equation embeddings. In one approach, they treated equations as tokens. In their second approach, they treated variables, symbols, and operators as tokens and equations as sequences of tokens. Equation embeddings were computed by taking the average of individual token embeddings. Mansouri et al. (2019) used a modified version of the latter method that extracted features from both the symbol layout tree and operator tree representations of an expression. Ahmed et al. (2021) used a complex schema where an equation was embedded using a combination of a message-passing network (to process its graph representation) and a residual neural network (to process its visual representation). Peng et al. (2021) trained BERT (Devlin et al., 2019) on mathematical datasets to create a pre-trained model for mathematical formula understanding.

Most of these approaches depend on embedding mathematical tokens and expressions using the surrounding textual information, effectively establishing a homomorphism between mathematical and textual information. While these approaches have produced crucial initial results, there are still two important limitations that need to be addressed. Firstly, an embedding scheme should be able to process equations without surrounding text, such as in the case of pure math texts like the Digital Library of Mathematical Functions (DLMF) (Lozier, 2003). Secondly, expressions may be written in a multitude of mathematically equivalent ways (consider $x^{-1} = \frac{1}{x}$ or $\sin(x) = \cos(x - \frac{\pi}{2})$). Embedding methods should recognize that such expressions are mathematically equivalent and should produce similar embeddings. Russin et al. (2021) and Schlag et al. (2019) have studied the representations learned by a Transformer model, trained on a math-reasoning dataset. Lample & Charton (2019) showed that their model generated multiple mathematically equivalent solutions for a first-order differential equation, showing a degree of semantic learning. Analogous to our approach, Zhang et al. (2017) trained a SEQ2SEQ model on sentence paraphrase pairs and used the last encoder hidden state as the vector representation of the input sentence.

Allamanis et al. (2017) and Liu (2022) have previously proposed EQNET and EQNET-L, respectively, for finding semantic representations of simple symbolic expressions. For EQNET, Allamanis et al. (2017) used the tree representation of a formula with a modified version of the recursive neural network (TreeNN). Their dataset is partitioned into equivalent classes, and they used this information while computing the training loss. They also introduced a regularization term in their objective, called subexpression autoencoder. EQNET-L (Liu, 2022) extended EQNET by adding a dropout layer and introducing a stacked version of the subexpression autoencoder. However, these approaches only focus on ensuring that the embeddings of *equivalent* expressions are grouped together. They do not consider or explore semantically similar but non-equivalent expressions. We define semantic similarity based on the operators present in expressions. For example, we consider $\sin(x + 1)$ to be more similar to $\sin(x)$ than $\log(x)$ or $\sinh(x + 1)$ (Section 5.2).

To this end, we look at mathematical expressions in isolation without any context and propose an approach to semantically represent these expressions in a continuous vector space. Our approach considers semantic similarity in addition to mathematical equivalence. Learning semantically rich representations of mathematical content would be useful in learning-based applications, like math problem-solving, as well as information retrieval.

## 3 Proposed Approach

We frame the problem of embedding mathematical expressions as a SEQ2SEQ learning problem and utilize the encoder-decoder framework. While natural language embedding approaches, like WORD2VEC, assume

that proximity in the text suggests similarity, we contend that for mathematical expressions, mathematical equivalence implies similarity. Furthermore, we hypothesize that if we train a SEQ2SEQ model to generate expressions that are mathematically equivalent to the input, the encoder would learn to generate embeddings that map semantically equivalent expressions together in the embedding vector space. Figure 1 shows an example of our approach. To accomplish this, we need a machine learning model capable of generating equivalent expressions and a dataset of equivalent expression pairs (Section 4).

**Model.** In encoder-decoder architectures, the encoder encodes the input sequence, and the decoder is conditioned on the encoded vector to generate an output sequence. In our approach, the encoder maps the input expression to a vector that is referred to as the *continuous vector representation* of the expression.

There are several choices for modeling encoders and decoders. We use the Transformer architecture (Vaswani et al., 2017) in this work. It has been successfully used in various NLP applications, like machine translation, language modeling, and summarization, to name a few, and it has also been shown to work with mathematical expressions (Lample & Charton, 2019).

**Embedding Vector.** Every encoder layer in the Transformer architecture generates hidden states corresponding to each token in the input sequence. We use max pooling on the hidden states of the last encoder layer to generate the continuous vector representation or the embedding vector of the input expression. The special tokens, like the start-of-expression and end-of-expression tokens, are not considered for max pooling. Furthermore, we considered the following other candidates to generate the embedding vectors but chose max pooling based on its performance in our initial experiments - (1) average pooling of the hidden states of the last encoder layer and (2) the hidden states of the last encoder layer corresponding to the first and the last tokens.

**Data Formatting.** Mathematical expressions are typically modeled as trees with nodes describing a variable or an operator (Ion et al., 1998). Since we are using a SEQ2SEQ model, we use the Polish (prefix) notation to convert a tree into a sequence of tokens. For example, $\frac{\sin(x)}{\cos(x)}$ is converted to the sequence $[\text{div}, \sin, x, \cos, x]$ (Figure 1). The primary reason for using the Polish notation is that we refer to the datasets created by Lample & Charton (2019) which are in Polish notation (details in Section 4). Furthermore, their work shows that the Transformer model is capable of generating valid prefix expressions without requiring constraints during decoding. In our experiments, we also find that invalid output prefix expressions become increasingly rare as the training progresses. Hence, we use the Polish notation for simplicity and leave the exploration of tree-based representations of mathematical expressions for future work. The tokens are encoded as one-hot vectors and passed through an embedding layer and positional encoding before being fed to the encoder or decoder.

**Decoding.** We use the beam search decoding (Koehn, 2004) to generate expressions at inference. The beam search algorithm is essential in finding an output with as high a probability as computationally possible. As the main focus of this work is vector representations and the beam search decoding is only used in equivalent expression generation, we use a simple version of beam search and do not employ length penalty, diverse beam search, or any other constraint during decoding.

## 4 Equivalent Expressions Dataset

Training our model requires a dataset of mathematically equivalent expression *pairs*. Working off a known collection of simple mathematical expressions, we use SymPy (Meurer et al., 2017) to create the *Equivalent Expressions Dataset*, consisting of ∼4.6M equivalent expression pairs. This dataset is available publicly on our project page.

**Generation.** To generate valid mathematical expressions, we refer to the work of Lample & Charton (2019). We take expressions from their publicly available datasets and perform pre-processing to remove the parts that are not necessary for our use case. This process gives us a set of valid mathematical expressions.

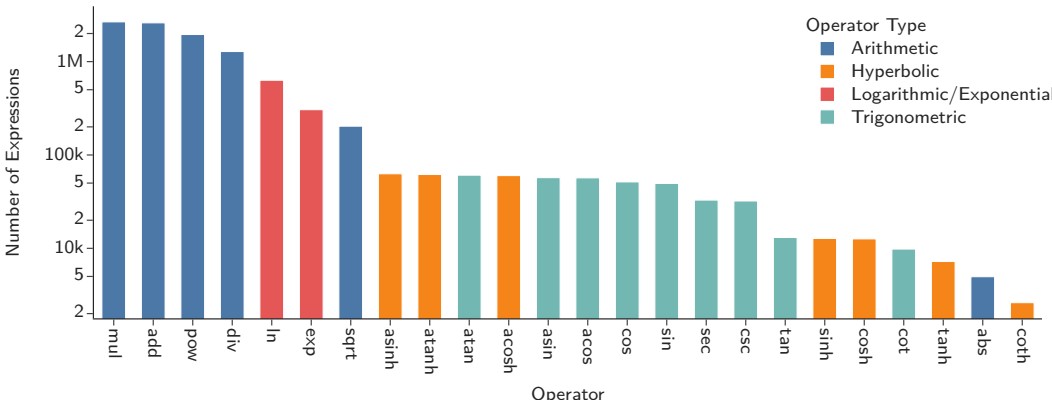

Figure 2: The number of expressions containing an operator in the Equivalent Expressions Dataset. Note that the dataset represents $x - y$ as $x + (-y)$, hence there is no subtraction operator in the plot.

Table 1: The number of operators and sequence lengths of training, validation, and test sets of the Equivalent Expressions Dataset.

| SET | # OPERATORS | SEQUENCE LENGTH |
|---|---|---|
| TRAINING | $5.68 \pm 1.32$ | $16.19 \pm 6.28$ |
| VALIDATION | $5.60 \pm 1.29$ | $15.03 \pm 4.31$ |
| TEST | $5.98 \pm 1.22$ | $16.20 \pm 4.13$ |

For these expressions, we use SymPy to generate mathematically equivalent but visually different counterparts (see Appendix A for details). For each pair of equivalent expressions $\mathbf{x}_1$ and $\mathbf{x}_2$, we add two examples $(\mathbf{x}_1, \mathbf{x}_2)$ and $(\mathbf{x}_2, \mathbf{x}_1)$ to the dataset. In our data generation process, we encounter expressions that result in NaN (Not a Number) when parsed using SymPy. We exclude examples that contain such expressions.

**Dataset.** Our training set has 4,662,300 input-output *pairs* comprised of 2,744,824 unique expressions. Note that the number of equivalent expression pairs per unique expression is not straightforward to compute because some expressions result in more equivalent expressions than others. All expressions are univariate and consist of the following operators:

- Arithmetic: $+, -, \times, /,$ abs, pow, sqrt.
- Trigonometric: sin, cos, tan, cot, sec, csc, $\sin^{-1}$ (asin), $\cos^{-1}$ (acos), $\tan^{-1}$ (atan).
- Hyperbolic: sinh, cosh, tanh, coth, $\sinh^{-1}$ (asinh), $\cosh^{-1}$ (acosh), $\tanh^{-1}$ (atanh).
- Logarithmic/Exponential: ln, exp.

The variable in the expressions is considered to be positive and real. For embedding analysis, this work only considers simple polynomial and transcendental mathematical expressions, and as such, we limit the input and output expressions to have a maximum of five operators. Figure 2 shows the number of expressions containing an operator. Note that one expression can contain multiple operators. The big disparity between the arithmetic and other operators arises because the arithmetic operators occur in almost all expressions. Other reasons for disparities are the ability of SymPy to generate equivalent expressions and our constraint of a maximum of five operators in each expression. Table 1 shows the number of operators in the training, validation, and test sets and corresponding sequence lengths.

Our validation and test sets contain a single expression per example instead of a pair. This is because there are multiple possible correct equivalent expressions for a given input expression. At inference, we use SymPy to determine if an output expression is equivalent to the input, instead of fixing one or multiple outputs for an input. The validation and test sets contain 2,000 and 5,000 expressions, respectively.

Note that an expression will only be present in one set, but an equivalent expression may be present in another set. For example, if $\sin(x)$ is present in the training set as either input or output, it will not be present in the validation and test sets. But $\cos(x - \frac{\pi}{2})$ may be present in either validation or test set, but not both.

## 5 Experiments

We consider two decoder settings for our experiments:

- Equivalent expression setting, or SEMEMB, in which the model is trained on disparate but mathematically equivalent expression pairs.
- Structural setting, or STRUCTEMB, in which the model is trained as an autoencoder to output the same expression as the input.

The proposed SEMEMB embeds an expression based on mathematical semantics, while STRUCTEMB only considers the visual layout of the expression. Though the focus of this work is to demonstrate the efficacy of SEMEMB, STRUCTEMB serves as an important contrast, demonstrating that SEMEMB yields representations that better describe *semantics* and are superior for clustering, retrieval, and analogy tasks.

### 5.1 Training Details

We use the Transformer architecture with 6 encoder layers, 6 decoder layers, 8 attention heads, and the ReLU activation. The model dimensions, indicated by $D_{\text{model}}$, are 512 and 64 for the experiments in Sections 5.2 and 5.3, respectively. The layer normalization is performed before other attention and feedforward operations. We use the Adam optimizer (Kingma & Ba, 2015) with a learning rate of $10^{-4}$ and do not consider expressions with more than 256 tokens. Refer to Appendix C for more details.

### 5.2 Embedding Evaluation

In this section, we evaluate the usefulness of the representations generated by the SEMEMB model. We train SEMEMB and STRUCTEMB on the Equivalent Expressions Dataset. As the dataset consists of equivalent expression pairs, we consider each expression as an independent example for training STRUCTEMB. The details and results of the training are presented in Section 5.4. For the experiments of this section, we disregard the decoder and use the encoder of the trained models to generate vector representations of input expressions.

Unlike the natural language embedding problem (Wang et al., 2018), there do not exist standard tests to quantitatively measure the embedding performance of methods built for mathematical expressions. Hence, our analysis in this section must be more qualitative in nature. These experiments show some interesting properties of the representations generated by SEMEMB and STRUCTEMB and demonstrate the efficacy of the proposed approach. The data used for the experiments are described in their respective sections.

**Embedding Plots.** To gauge if similar expressions are clustered in the embedding vector space, we plot and compare the representations generated by the SEMEMB and STRUCTEMB models. For this experiment, we use a set of 7,000 simple expressions belonging to four categories: hyperbolic, trigonometric, polynomial, and logarithmic/exponential. Each expression is either a polynomial or contains hyperbolic, trigonometric, logarithmic, or exponential operators. Each expression contains only one category of operators. 35% of these expressions are from the training set and the remaining 65% are unseen expressions. Below are a few examples of expressions belonging to each class:

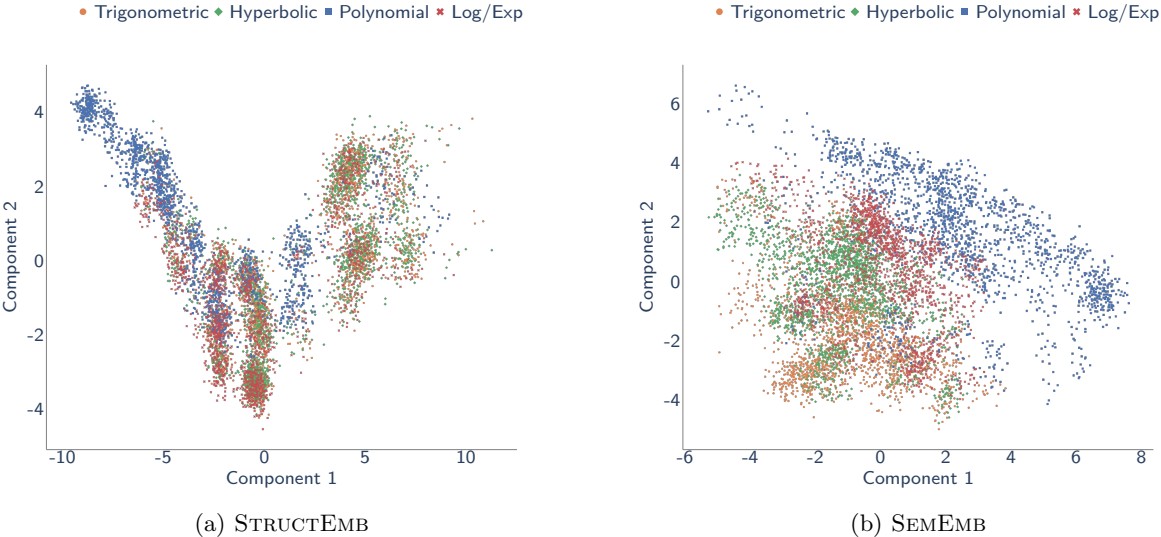

Figure 3: Plots for the embedding vectors generated by STRUCTEMB (autoencoder) and SEMEMB (proposed equivalent-expression method). PCA is used to reduce the dimensionality from 512 to 2. SEMEMB groups expressions with operators of the same type together, indicating its ability to understand semantics, whereas STRUCTEMB groups expressions mainly based on their visual structure. The interactive versions of these plots are available on our project page.

- Polynomial: $x^2 + 2x + 5$, $2x + 2$
- Trigonometric: $\sin(x)\tan(x)$, $\cos^5(4x)$
- Hyperbolic: $\cosh(x - 4)$, $\sinh(x\cos(2))$
- Log/Exp: $e^{-2x-4}$, $\log(x + 3)^3$

We use Principal Component Analysis (PCA) for dimensionality reduction. Figure 3 shows the plots for STRUCTEMB and SEMEMB. We observe from these plots that the clusters in the SEMEMB plot are more distinguishable compared to the STRUCTEMB plot. SEMEMB does a better job at grouping similar expressions together, suggesting its ability to understand semantics. For SEMEMB, there is an overlap between expressions belonging to hyperbolic and logarithmic/exponential classes. This is expected because hyperbolic operators can be written in terms of the exponential operator. Furthermore, STRUCTEMB focuses on the overall token similarity. For example, representations generated by STRUCTEMB for $\tan\left(x\frac{\sqrt{2}}{2}\right)$, $x^2(x^2 - x)$, $\sinh^{-1}\left(x\frac{\sqrt{2}}{2}\right)$, and $\log\left(x\frac{\sqrt{2}}{2}\right)$ are grouped together. On the other hand, representations generated by SEMEMB capture semantics in addition to the syntactic structure.

**Distance Analysis.** To understand the applicability of the proposed approach for the information retrieval task, we perform distance analysis on the embeddings generated by STRUCTEMB and SEMEMB. We use all expressions from the training set of the Equivalent Expressions Dataset as the pool of expressions for this experiment. The similarity between two expressions is defined as the inverse of the cosine distance between their embedding vectors. We find the five closest expressions to a given query expression. Table 2 shows the results of this experiment. We observe that the closest expressions computed using SEMEMB are more similar to the query in terms of the syntactic structure and operators. On the other hand, STRUCTEMB focuses on the overall token similarity. For example, the first query in Table 2 consists of polynomial and trigonometric expressions. The closest expressions computed using SEMEMB follow the same structure, whereas STRUCTEMB seems to put more emphasis on the polynomial multiplier. This behavior is also apparent in the second example. We believe that the ability of SEMEMB to group similar expressions, in

Table 2: Expressions closest to a given query based on the representations generated by STRUCTEMB and SEMEMB. It is evident that SEMEMB does a better job at learning semantics and the overall structure of the expressions. Refer to Appendix E for more examples.

| QUERY | STRUCTEMB | SEMEMB |
|---|---|---|
| $4x^2 \cos{(3x-1)}$ | $4x^2 e^{-3x-1}$ | $-10x^2 \cos{(x-10)}$ |
| | $4x^2 e^{2x-1}$ | $-10x^2 \cos{(x+10)}$ |
| | $4x^2 e^{-2x-1}$ | $x^2 \cos{(x-10)}$ |
| | $4x^2 e^{-3x-9}$ | $x^2 \cos{(x+1)}$ |
| | $4x^2 e^{-x-1}$ | $x^2 \cos{(x-30)}$ |
| $x^2 + \log{(x)} + 1$ | $\sqrt{x} + x^2 + \log{(x)}$ | $x^2 + \log{(x)} + 1$ |
| | $\sqrt{x} + x^2 + e^x$ | $x^2 + \log{(x)} + 3$ |
| | $\sqrt{x} + x^2 + \mathrm{acos}{(x)}$ | $x^2 + \log{(x)} + 2$ |
| | $\log{\left(x^{x^2}\right)} + \sinh{(x)}$ | $x^2 + x + \log{(x)} + 1$ |
| | $\log{\left(x^{x^2}\right)} + \tan{(x)}$ | $x^2 + x + \log{(x)}$ |

terms of operators and the syntactic structure, can prove useful in information retrieval problems, where the aim is to find similar expressions to a given query. Refer to Appendix E for more examples.

To quantify the notion of similarity, we use the tree edit distance between the operator trees of two expressions.[2] We consider 2,000 query expressions and analyze the most similar expression returned by STRUCTEMB and SEMEMB for a given query. A cost of one unit is used for inserting, deleting, or updating a token while computing the distance. The tree edit distance between the query expression and the result shows the following:

- *Scenario 1.* If we compare expressions in their original form, STRUCTEMB returns an expression strictly closer to the query in 1,044 cases and SEMEMB in 261 cases.

- *Scenario 2.* When we exclude constant multipliers and additions, i.e., we consider $af(x) + b$ and $f(x)$ to be at a distance of zero, SEMEMB returns an expression strictly closer to the query in 683 cases and STRUCTEMB in 402 cases. Note that $a$ and $b$ could be simple constants, like 1, $\sqrt{2}$, etc or constant expressions, like $\cos(1)$, $\cos(\log(1))$, etc.

It is interesting to see the change in results as we move from the first scenario to the second one. These results indicate that SEMEMB puts more emphasis on operators compared to STRUCTEMB, and STRUCTEMB considers the overall token similarity. The five closest expressions to the query show a similar trend.

**Embedding Algebra.** Word embeddings generated by methods like WORD2VEC (Mikolov et al., 2013) and GloVe (Pennington et al., 2014) exhibit an interesting property that simple algebraic operations on the embedding vectors can be used to solve analogies of the form "$x_1$ is to $y_1$ as $x_2$ is to $y_2$". Following along the same lines, we perform simple algebraic operations on the embeddings generated by STRUCTEMB and SEMEMB. For a given triplet of expressions $x_1$, $y_1$, and $y_2$, we compute

$$z = \mathrm{emb}(x_1) - \mathrm{emb}(y_1) + \mathrm{emb}(y_2) \tag{1}$$

where "emb" represents a function that returns the embedding vector of an input expression. We then find an expression with the embedding vector closest to $z$ in terms of cosine similarity, excluding $x_1$, $y_1$, and $y_2$.

To create a pool of expressions for this experiment, we use all expressions from the training set and add any missing expressions that are required for an analogy. We create 20 analogy examples, 12 of which are based on mathematical identities, and the remaining are based on simple substitutions. Table 3 shows the results for STRUCTEMB and SEMEMB. It is interesting to observe that SEMEMB works for nine examples that are

---

[2]Implementation in Python: https://pythonhosted.org/zss/.

Table 3: Examples of embedding algebra on the representations generated by STRUCTEMB and SEMEMB. The correct predictions are shown in **bold**.

| $x_1$ | $y_1$ | $y_2$ | $x_2$ (EXPECTED) | $x_2$ (PREDICTED) | |
| --- | --- | --- | --- | --- | --- |
| | | | | STRUCTEMB | SEMEMB |
| $\cos(x)$ | $\sin(x)$ | $\csc(x)$ | $\sec(x)$ | $\mathbf{sec\,(x)}$ | $\mathbf{sec\,(x)}$ |
| $\cos(x)$ | $\sin(x)$ | $\cot(x)$ | $\tan(x)$ | $\sqrt{x}$ | $x + \cot(x)$ |
| $\sin(x)$ | $\cos(x)$ | $\cosh(x)$ | $\sinh(x)$ | $\tanh(x)$ | $\mathbf{sinh\,(x)}$ |
| $\sin(x)$ | $\csc(x)$ | $\sec(x)$ | $\cos(x)$ | $\mathbf{cos\,(x)}$ | $\mathbf{cos\,(x)}$ |
| $\sin(x)$ | $\csc(x)$ | $\cot(x)$ | $\tan(x)$ | $\cos(x)$ | $\mathbf{tan\,(x)}$ |
| $\sin(-x)$ | $-\sin(x)$ | $\cos(x)$ | $\cos(-x)$ | $\sin(x)$ | $\mathbf{cos\,(-x)}$ |
| $\tan(-x)$ | $-\tan(x)$ | $-\cot(x)$ | $\cot(-x)$ | $\sinh(-x)$ | $\mathbf{cot\,(-x)}$ |
| $\sin(-x)$ | $-\sin(x)$ | $-\cot(x)$ | $\cot(-x)$ | $\tan(-x)$ | $\mathbf{cot\,(-x)}$ |
| $\sinh(-x)$ | $-\sinh(x)$ | $\cosh(x)$ | $\cosh(-x)$ | $\text{acosh}(x)$ | $\mathbf{cosh\,(-x)}$ |
| $\sinh(-x)$ | $-\sinh(x)$ | $-\tanh(x)$ | $\tanh(-x)$ | $\mathbf{tanh\,(-x)}$ | $\mathbf{tanh\,(-x)}$ |
| $\sinh(-x)$ | $-\sinh(x)$ | $-\cosh(x)$ | $-\cosh(-x)$ | $\tan(-x)$ | $\cosh(-x)$ |
| $\sinh(-x)$ | $-\sinh(x)$ | $\tanh(x)$ | $-\tanh(-x)$ | $\text{acosh}(x)$ | $\tanh(-x)$ |
| $x^2$ | $x$ | $\sin(x)$ | $\sin^2(x)$ | $\mathbf{sin^2\,(x)}$ | $x^2 + \sin(x)$ |
| $x^2$ | $x$ | $\cos(x)$ | $\cos^2(x)$ | $\mathbf{cos^2\,(x)}$ | $\mathbf{cos^2\,(x)}$ |
| $x^2$ | $x$ | $\log(x)$ | $\log(x)^2$ | $\log(\sqrt{2})$ | $\log(x^2)$ |
| $x^2$ | $x$ | $e^x$ | $(e^x)^2$ | $x^4$ | $x^2 e^x$ |
| $e^x$ | $x$ | $2x$ | $e^{2x}$ | $\mathbf{e^{2x}}$ | $2e^x$ |
| $x^2$ | $x$ | $x+2$ | $(x+2)^2$ | $\tan^2(x)$ | $x^2 + 2$ |
| $\log(x)$ | $x$ | $\sin(x)$ | $\log(\sin(x))$ | $\mathbf{log\,(sin\,(x))}$ | $\sin(\log(x))$ |
| $\sin(x)$ | $x$ | $\log(x)$ | $\sin(\log(x))$ | $\log(\sin(x))$ | $\mathbf{sin\,(log\,(x))}$ |

based on mathematical identities. It demonstrates a degree of semantic learning. STRUCTEMB performs poorly for the identity-based examples and gets four substitution-based examples right, demonstrating that STRUCTEMB understands the visual structure to a certain extent. The results of this analysis further bolster the efficacy of SEMEMB for learning semantically rich embeddings.

### 5.3 Comparison with Existing Methods

As discussed in Section 2, prior works (Allamanis et al., 2017; Liu, 2022) have examined if machine learning models can generate similar representations of mathematically equivalent expressions. It should be noted that these prior works only focus on generating similar representations for mathematically equivalent expressions and do not explore the applicability of their methods to the similarity of non-equivalent expressions. But this historical perspective serves as an established evaluation of the representations generated by SEMEMB and STRUCTEMB.

If we refer to all mathematically equivalent expressions as belonging to a class, this property is measured as the proportion of $k$ nearest neighbors of each test expression that belong to the same class (Allamanis et al., 2017). For a test expression $q$ belonging to a class $c$, the score is defined as

$$score_k(q) = \frac{|\mathbb{N}_k(q) \cap c|}{\min(k, |c|)} \tag{2}$$

where $\mathbb{N}_k(q)$ represents $k$ nearest neighbors of $q$ based on cosine similarity.

We use the datasets published by Allamanis et al. (2017) for this evaluation. These datasets contain equivalent expressions from the Boolean (BOOL) and polynomial (POLY) domains. In these datasets, a class is defined by an expression, and all the equivalent expressions belong to the same class. The datasets are split into training, validation, and test sets. The expressions in the training and validation sets come from the same classes. The dataset contains two test sets: (1) SEENEQCLASS containing classes that are present in the training set, and (2) UNSEENEQCLASS containing classes that are not present in the training set (to measure the generalization to the unseen classes).

Table 4: $score_5(\%)$ on UNSEENEQCLASS of the SEMVEC datasets. The scores for EQNET and EQNET-L are from their published work (Allamanis et al., 2017; Liu, 2022).

| DATASET | EQNET $score_5(\%)$ | EQNET-L $score_5(\%)$ | STRUCTEMB $score_5(\%)$ | TRAINING SET SIZE | SEMEMB $score_5(\%)$ | TRAINING SET SIZE |
|---|---|---|---|---|---|---|
| BOOL8 | 58.1 | - | 30.6 | 146,488 | 100.0 | 16,143,072 |
| ONEV-POLY13 | 90.4 | - | 38.7 | 60,128 | 99.6 | 9,958,406 |
| SIMPPOLY10 | 99.3 | - | 40.1 | 31,143 | 99.8 | 6,731,858 |
| SIMPBOOL8 | 97.4 | - | 36.9 | 21,604 | 99.4 | 4,440,450 |
| BOOL10 | 71.4 | - | 10.8 | 25,560 | 91.3 | 3,041,640 |
| SIMPBOOL10 | 99.1 | - | 24.4 | 13,081 | 95.5 | 1,448,804 |
| BOOLL5 | 75.2 | - | 38.3 | 23,219 | 36.0 | 552,642 |
| POLY8 | 86.2 | 87.1 | 32.7 | 6,785 | 87.3 | 257,190 |
| SIMPPOLY8 | 98.9 | 98.0 | 47.6 | 1,934 | 98.9 | 113,660 |
| SIMPBOOLL5 | 85.0 | 72.1 | 55.1 | 6,009 | 71.1 | 66,876 |
| ONEV-POLY10 | 81.3 | 80.0 | 59.8 | 767 | 74.1 | 25,590 |
| BOOL5 | 65.8 | 73.7 | 36.4 | 712 | 57.7 | 17,934 |
| POLY5 | 55.3 | - | 5.7 | 352 | 45.2 | 1,350 |
| SIMPPOLY5 | 65.6 | 56.3 | 28.1 | 156 | 20.8 | 882 |

Our approach requires data to be in the input-output format. For SEMEMB, the input and output are mathematically equivalent expressions. To transform the training set into the input-output format for SEMEMB, we generate all possible pairs for the expressions belonging to the same class. To limit the size of the generated training set, we select a maximum of 100,000 random pairs from each class. For STRUCTEMB, the input and output are the same expressions. For this setting, we treat each expression present in the training set as an independent example. Table 4 shows the training set sizes obtained by these transformations for SEMEMB and STRUCTEMB. We use the validation and test sets in their original form.

As mentioned in Section 5.1, both STRUCTEMB and SEMEMB are trained with a model dimensionality of 64. To enable a fair comparison with the existing approaches, we use the same model configuration and hyperparameter values for all 14 SEMVEC datasets. Refer to Appendix C for a detailed description of the model configuration and hyperparameter values.

For evaluating our models, we use the UNSEENEQCLASS test set. Table 4 shows the scores achieved by our approach. We observe that the representations learned by SEMEMB capture semantics and not just the syntactic structure. It should be noted that our approach does not use the distance between representations at the time of training and only trains the model to generate equivalent expressions, whereas the training for EQNET and EQNET-L explicitly pushes the representations of equivalent expressions closer in the embedding vector space. We further observe that SEMEMB performs better than the existing approaches on the datasets with a sufficiently high number of training examples. For other datasets, the model does not perform well on UNSEENEQCLASS but does well on SEENEQCLASS and the validation set (See Appendix D for details). We believe that this is because of not having enough examples in the training set for generalization to the unseen classes, and these datasets may perform better with simpler SEQ2SEQ models and different hyperparameters. We leave this exploration for future work.

In our experiments, STRUCTEMB does not perform as well as SEMEMB. One possible reason for this behavior may be the less number of examples for training the model in the autoencoder setting. To rule out this possibility, we train models with three model dimensions of 32, 64, and 128 with 179K, 703K, and 2.8M parameters, respectively. We do not observe any significant difference in the performance across these model dimensions. The model also does not achieve good scores on the training and validation sets (See Appendix D for details). Furthermore, SEMEMB performs better than STRUCTEMB on the datasets with comparable training set sizes, for example, consider STRUCTEMB on BOOL8 vs SEMEMB on SIMPPOLY8. With these observations, we conclude with good certainty that the representations learned by STRUCTEMB are inferior to those learned by SEMEMB in terms of capturing semantics.

Table 5: Accuracy of STRUCTEMB and SEMEMB on the Equivalent Expressions Dataset. STRUCTEMB has an easier task of generating the input at the decoder, while SEMEMB must generate a mathematically equivalent expression.

| BEAM SIZE | STRUCTEMB | SEMEMB |
|---|---|---|
| 1 | 0.9996 | 0.7206 |
| 10 | 0.9998 | 0.8118 |
| 50 | 0.9998 | 0.8386 |

### 5.4 Training Efficacy

This section presents the efficacy of training SEMEMB and STRUCTEMB models on the Equivalent Expressions Dataset that are used in Section 5.2. It is important to emphasize that SEMEMB and STRUCTEMB are trained on fundamentally different problems. STRUCTEMB is an autoencoder and is trained to generate the input expression exactly, while SEMEMB is trained to generate visually different but mathematically equivalent expressions. We show that SEMEMB can learn to generate mathematically equivalent expressions for a given input. To evaluate if two expressions $\mathbf{x}_1$ and $\mathbf{x}_2$ are mathematically equivalent, we simplify their difference $\mathbf{x}_1 - \mathbf{x}_2$ using SymPy and compare it to 0. In this setting, if the model produces an expression that is the same as the input, we do not count it as a model success. There are instances in which SymPy takes a significant time to simplify an expression and eventually fails with out-of-memory errors. To handle these cases, we put a threshold on its execution time. If the simplification operation takes more time than the threshold, we count it as a model failure. At inference, we use the beam search algorithm to generate output expressions. As an output can be verified programmatically, we consider all outputs in a beam. If any of the outputs are correct, we count it as a model success.

The accuracy of these models is shown in Table 5. First, we note that STRUCTEMB is easily able to encode and generate the input expression and achieves a near-perfect accuracy with greedy decoding (beam size = 1). Second, the SEMEMB results demonstrate that generating mathematically equivalent expressions is a significantly harder task. For this setting, we observe an improvement of 11% with a beam size of 50. However, we also see an increase in the number of invalid prefix expressions being assigned high log probabilities with this beam size. This experiment demonstrates that both STRUCTEMB and SEMEMB are capable of generating mathematical expressions, and SEMEMB can learn to generate expressions that are mathematically equivalent to the input expression. These results also indicate that it is harder to train a model to generate mathematically equivalent expressions compared to training it as an autoencoder.

## 6 Conclusion

In this work, we developed a framework to represent mathematical expressions in a continuous vector space. We showed that a SEQ2SEQ model could be trained to generate expressions that are mathematically equivalent to the input. The encoder of this model could be used to generate vector representations of mathematical expressions. We performed quantitative and qualitative experiments and demonstrated that these representations were semantically rich. Our experiments also showed that these representations were better at grouping similar expressions and performed better on the analogy task than the representations generated by an autoencoder.

We also note certain limitations. There is a need for downstream tasks or better evaluation metrics to evaluate the quality of the learned representations of mathematical expressions. Though we evaluated our approach on the SEMVEC datasets, we could not perform a similar evaluation on the Equivalent Expressions Dataset due to a limited number of equivalent expressions per expression. Furthermore, the notion of similarity is not well-defined for mathematical expressions. Though the analysis with the tree edit distance shows an interesting contrast between STRUCTEMB and SEMEMB, it only looks at tokens and does not consider semantics. More work is needed to establish a comprehensive and quantifiable measure of similarity for mathematical expressions. Lastly, the dataset presented in this work consists of univariate expressions

that contain simple algebraic and transcendental operators. It may be extended to include expressions with multiple variables and more complex operators. We leave these avenues for future research.

**Acknowledgments**

We thank Prof. Shaloo Rakheja (University of Illinois Urbana-Champaign) for providing computing resources for this work. This work also utilizes resources supported by The Grainger College of Engineering, the National Science Foundation's Major Research Instrumentation program, grant #1725729, and the University of Illinois Urbana-Champaign. Finally, we thank the anonymous reviewers and the action editor at TMLR for insightful comments and feedback, which helped us improve this work.

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

## A   Equivalent Expression Generation

We use SymPy to generate mathematically equivalent expressions for a given formula. We apply the operations shown in Table 6 to get mathematically equivalent expressions. The examples shown in the table are intended to give an idea about each function. The actual behavior of these functions is much more complex. Refer to the SymPy documentation for more details.[3] We use the `REWRITE` function for expressions containing trigonometric or hyperbolic operators. Specifically, we use this function to rewrite an expression containing trigonometric (or hyperbolic) operators in terms of other trigonometric (or hyperbolic) operators.

Table 6: List of SymPy functions with examples that are used to generate equivalent expressions.

| FUNCTION | INPUT | OUTPUT |
|---|---|---|
| SIMPLIFY | $\sin(x)^2 + \cos(x)^2$ | $1$ |
| EXPAND | $(x+1)^2$ | $x^2 + 2x + 1$ |
| FACTOR | $x^2 + 5x + 6$ | $(x+2)(x+3)$ |
| CANCEL | $(x^3 + 2x)/x$ | $x^2 + 2$ |
| TRIGSIMP | $\sin(x)\cot(x)$ | $\cos(x)$ |
| EXPAND_LOG | $\ln(x^2)$ | $2\ln(x)$ |
| LOGCOMBINE | $\ln(x) + \ln(2)$ | $\ln(2x)$ |
| REWRITE | $\sin(x),\ \cos$ | $\cos(x - \pi/2)$ |
|  | $\sinh(x),\ \tanh$ | $2\tanh\left(\frac{x}{2}\right)(1 - \tanh^2\left(\frac{x}{2}\right))^{-1}$ |

## B   Equivalent Expressions Dataset

The expressions in the Equivalent Expressions Dataset may contain operators from multiple operator types (shown in Table 1). For example, 1,424,243 expressions contain operators from a single operator type, 1,186,808 expressions contain two types of operators, 130,547 expressions contain three types of operators, and so on. Table 7 shows a few examples from the Equivalent Expressions Dataset.

## C   Training Details

We use the PyTorch implementation of the Transformer architecture for our experiments with a modified version of the decoder to enable caching to speed up the generation process at inference.[4] We use 6 encoder layers, 6 decoder layers, 8 attention heads, 0.1 dropout probability, and the ReLU activation for our experiments. The layer normalization is performed in the encoder and decoder layers before other attention and feedforward operations. We use the Adam optimizer with a learning rate of $10^{-4}$, a fixed seed of 42 for reproducibility, and a label smoothing of 0.1 while computing the loss.

For the Equivalent Expressions Dataset, we use a model dimension of 512, a feedforward dimension of 2048, and a batch size of 256. These experiments were run on two 32GB V100 GPUs. We use early stopping with 300K minimum steps, 1M maximum steps, and patience of 30K steps. In terms of epochs, this translates to 17 minimum epochs, 55 maximum epochs, and patience of 2 epochs for SEMEMB and 28 minimum epochs, 94 maximum epochs, and patience of 3 epochs for STRUCTEMB. We evaluate the model at the end of every epoch.

For the SEMVEC datasets, we use a model dimension of 64, a feedforward dimension of 256, and a batch size of 512. These experiments were run on one 16GB V100 GPU. We use early stopping with 50K minimum steps, 1M maximum steps, and patience of 20K steps or 2 epochs (whichever is smaller). Table 8 shows these values in terms of epochs for different datasets. We evaluate the model at the end of every epoch.

---

[3]https://docs.sympy.org/latest/tutorial/simplification.html
[4]https://pytorch.org/docs/1.12/generated/torch.nn.Transformer.html

Table 7: Examples from the Equivalent Expressions Dataset.

| INPUT | OUTPUT |
|---|---|
| $1 \cdot \frac{1}{2} x^{\frac{5}{2}} - \frac{5x^{\frac{3}{2}}}{2}$ | $1 \cdot \frac{1}{2} x^{\frac{3}{2}} (x - 5)$ |
| $1 \cdot \frac{1}{2} x \log(x) + 2x + \frac{5}{2}$ | $1 \cdot \frac{1}{2} (x \log(x) + 4x + 5)$ |
| $x + e^x - 20 + \frac{1}{\sec\left(x - \frac{\pi}{2}\right)}$ | $x + e^x + \sin(x) - 20$ |
| $x^2 (\text{acos}(x) + 1) + x + 3$ | $x(x\,\text{acos}(x) + x) + x + 3$ |
| $\frac{5\pi x^4}{4} + 1 \cdot \frac{1}{4} \pi x$ | $1 \cdot \frac{1}{4} \pi \left(5x^4 + x\right)$ |
| $x + \log\left(x^2\right) - 1 + 2\log(5)$ | $x + \log\left(25x^2\right) - 1$ |
| $5e^x + \log\left(x^{\frac{5}{2}}\right)$ | $\frac{5 \cdot (2e^x + \log(x))}{2}$ |
| $-3x\,\text{acos}(x) + 4x$ | $3x(1 - \text{acos}(x)) + x$ |
| $2(\text{atanh}(x) + 1)\log(2)$ | $(\text{atanh}(x) + 1)\log(4)$ |
| $-\frac{10x}{\coth(\sin(10))} + 10$ | $-10(x\tanh(\sin(10)) - 1)$ |
| $-20\sin(x) + 20\cosh(x)$ | $-20\cos\left(x - \frac{\pi}{2}\right) + 20\cosh(x)$ |
| $(-x + \text{asinh}(3))(x + 1)$ | $-(x + 1)(x - \text{asinh}(3))$ |
| $x\cos(x) + \text{acosh}(x)$ | $x\sin\left(x + 1 \cdot \frac{1}{2}\pi\right) + \text{acosh}(x)$ |
| $1 \cdot \frac{1}{4} x + 1 \cdot \frac{1}{4} + 1 \cdot \frac{1}{4} \cdot \frac{1}{x} \log(x)$ | $1 \cdot \frac{1}{4} \cdot \frac{1}{x}\left(x^2 + x + \log(x)\right)$ |
| $x\,\text{atan}(3) + 1 \cdot \frac{1}{2}\,\text{atan}(3)$ | $\left(x + 1 \cdot \frac{1}{2}\right)\text{atan}(3)$ |

Table 8: The minimum, maximum, and patience epochs for early stopping used in the SEMVEC training.

| DATASET | STRUCTEMB | | | SEMEMB | | |
|---|---|---|---|---|---|---|
| | MINIMUM | MAXIMUM | PATIENCE | MINIMUM | MAXIMUM | PATIENCE |
| BOOL8 | 175 | 3,485 | 70 | 2 | 32 | 2 |
| ONEV-POLY13 | 424 | 8,475 | 170 | 3 | 52 | 2 |
| SIMPPOLY10 | 820 | 16,394 | 328 | 4 | 77 | 2 |
| SIMPBOOL8 | 1,163 | 23,256 | 466 | 6 | 116 | 3 |
| BOOL10 | 1,000 | 20,000 | 400 | 9 | 169 | 4 |
| SIMPBOOL10 | 1,924 | 38,462 | 770 | 18 | 354 | 8 |
| BOOLL5 | 1,087 | 21,740 | 435 | 47 | 926 | 19 |
| POLY8 | 3,572 | 71,429 | 1,429 | 100 | 1,989 | 40 |
| SIMPPOLY8 | 12,500 | 250,000 | 5,000 | 226 | 4,505 | 91 |
| SIMPBOOLL5 | 4,167 | 83,334 | 1,667 | 382 | 7,634 | 153 |
| ONEV-POLY10 | 25,000 | 500,000 | 10,000 | 1,000 | 20,000 | 400 |
| BOOL5 | 25,000 | 500,000 | 10,000 | 1,389 | 27,778 | 556 |
| POLY5 | 50,000 | 1,000,000 | 20,000 | 16,667 | 333,334 | 6,667 |
| SIMPPOLY5 | 50,000 | 1,000,000 | 20,000 | 25,000 | 500,000 | 10,000 |

## D    Results for the SemVec Datasets

To compute $score_k(q)$ for our model, we use the source code provided by Allamanis et al. (2017).[5]

Tables 9 and 10 show the scores achieved by STRUCTEMB and SEMEMB on the validation and SEENEQCLASS test sets of the SEMVEC datasets, respectively. For STRUCTEMB, we try three model dimensions of 32, 64,

---

[5]https://github.com/mast-group/eqnet

Table 9: $score_5(\%)$ achieved by STRUCTEMB and SEMEMB on the validation sets of the SEMVEC datasets.

| DATASET | STRUCTEMB | | | SEMEMB |
|---|---|---|---|---|
| | $D_{\text{MODEL}} = 32$ | $D_{\text{MODEL}} = 64$ | $D_{\text{MODEL}} = 128$ | |
| BOOL8 | 36.2 | 36.3 | 35.5 | 100.0 |
| ONEV-POLY13 | 38.3 | 39.0 | 38.5 | 99.9 |
| SIMPPOLY10 | 37.5 | 31.5 | 34.0 | 99.9 |
| SIMPBOOL8 | 42.3 | 46.7 | 44.5 | 99.8 |
| BOOL10 | 9.6 | 9.9 | 9.1 | 93.9 |
| SIMPBOOL10 | 23.2 | 26.6 | 27.2 | 96.3 |
| BOOLL5 | 47.4 | 47.6 | 47.7 | 98.2 |
| POLY8 | 36.4 | 36.5 | 36.0 | 98.6 |
| SIMPPOLY8 | 36.5 | 35.4 | 34.6 | 99.8 |
| SIMPBOOLL5 | 75.0 | 76.1 | 74.4 | 99.5 |
| ONEV-POLY10 | 42.1 | 46.3 | 42.5 | 83.0 |
| BOOL5 | 42.4 | 42.1 | 35.4 | 81.0 |
| POLY5 | 11.6 | 9.5 | 7.4 | 43.2 |
| SIMPPOLY5 | 20.0 | 28.8 | 32.5 | 43.8 |

Table 10: $score_5(\%)$ achieved by STRUCTEMB and SEMEMB on SEENEQCLASS of the SEMVEC datasets.

| DATASET | STRUCTEMB | | | SEMEMB |
|---|---|---|---|---|
| | $D_{\text{MODEL}} = 32$ | $D_{\text{MODEL}} = 64$ | $D_{\text{MODEL}} = 128$ | |
| BOOL8 | 36.1 | 36.1 | 35.5 | 100.0 |
| ONEV-POLY13 | 38.0 | 38.5 | 38.1 | 99.9 |
| SIMPPOLY10 | 37.7 | 32.2 | 34.5 | 99.9 |
| SIMPBOOL8 | 41.6 | 46.1 | 43.5 | 99.8 |
| BOOL10 | 9.6 | 9.8 | 8.9 | 94.0 |
| SIMPBOOL10 | 23.0 | 26.8 | 27.3 | 96.5 |
| BOOLL5 | 40.5 | 42.0 | 42.2 | 68.7 |
| POLY8 | 34.9 | 35.0 | 34.5 | 95.2 |
| SIMPPOLY8 | 38.5 | 36.7 | 36.1 | 99.6 |
| SIMPBOOLL5 | 58.2 | 59.3 | 58.3 | 80.9 |
| ONEV-POLY10 | 36.1 | 39.6 | 37.9 | 77.3 |
| BOOL5 | 41.1 | 43.2 | 34.6 | 80.2 |
| POLY5 | 14.7 | 8.2 | 4.2 | 46.2 |
| SIMPPOLY5 | 30.9 | 32.6 | 42.2 | 47.9 |

and 128 to rule out the effect of underfitting and overfitting while training the model. Table 11 shows the scores achieved by these models on UNSEENEQCLASS. It can be seen that changing the number of parameters in the model does not have a significant effect on the scores.

We also perform an evaluation of compositionality on the SEMVEC datasets (Allamanis et al., 2017). For this evaluation, we train our model on a simpler dataset and evaluate it on a complex dataset. For example, a model trained on BOOL5 is evaluated on BOOL10. We use UNSEENEQCLASS for the evaluation. The results of this experiment are shown in Table 12.

# E    Distance Analysis

Table 14 shows more examples of the distance analysis as discussed in Section 5.2.

Table 11: $score_5$(%) achieved by STRUCTEMB on UNSEENEQCLASS of the SEMVEC datasets for different model dimensions.

| DATASET | $D_{\text{MODEL}} = 32$ | $D_{\text{MODEL}} = 64$ | $D_{\text{MODEL}} = 128$ |
|---|---|---|---|
| BOOL8 | 31.1 | 30.6 | 25.9 |
| ONEV-POLY13 | 37.7 | 38.7 | 38.4 |
| SIMPPOLY10 | 47.4 | 40.1 | 44.9 |
| SIMPBOOL8 | 33.1 | 36.9 | 34.1 |
| BOOL10 | 10.7 | 10.8 | 8.6 |
| SIMPBOOL10 | 20.5 | 24.4 | 25.4 |
| BOOLL5 | 35.6 | 38.3 | 38.2 |
| POLY8 | 32.2 | 32.7 | 32.1 |
| SIMPPOLY8 | 50.4 | 47.6 | 47.3 |
| SIMPBOOLL5 | 55.2 | 55.1 | 54.9 |
| ONEV-POLY10 | 58.3 | 59.8 | 59.7 |
| BOOL5 | 36.7 | 36.4 | 28.1 |
| POLY5 | 14.9 | 5.7 | 6.6 |
| SIMPPOLY5 | 18.8 | 28.1 | 17.7 |

Table 12: Compositionality test of SEMEMB. $score_5$(%) achieved on UNSEENEQCLASS of a SEMVEC dataset represented by TO by a model trained on the dataset represented by FROM. For scores achieved by EQNET, refer to Figure 7 in Allamanis et al. (2017).

| FROM → TO | $score_5$(%) |
|---|---|
| BOOL5 → BOOL10 | 12.5 |
| BOOL5 → BOOL8 | 32.5 |
| BOOLL5 → BOOL8 | 29.3 |
| ONEV-POLY10 → ONEV-POLY13 | 29.8 |
| POLY5 → POLY8 | 32.9 |
| POLY5 → SIMPPOLY10 | 45.1 |
| POLY8 → ONEV-POLY13 | 29.9 |
| POLY8 → SIMPPOLY10 | 58.9 |
| POLY8 → SIMPPOLY8 | 98.9 |
| SIMPPOLY5 → SIMPPOLY10 | 45.5 |
| SIMPPOLY8 → SIMPPOLY10 | 65.9 |

## F Equivalent Expression Generation

For the Equivalent Expressions Dataset, Table 13 shows the accuracy of STRUCTEMB and SEMEMB on the validation set of the Equivalent Expressions Dataset with beam sizes of 1, 10, and 50.

Table 13: Accuracy of STRUCTEMB and SEMEMB on the validation set of the Equivalent Expressions Dataset.

| BEAM SIZE | STRUCTEMB | SEMEMB |
|---|---|---|
| 1 | 0.9995 | 0.7615 |
| 10 | 0.9995 | 0.8365 |
| 50 | 0.9995 | 0.8665 |

Table 14: Expressions closest to a given query computed based on embeddings generated by STRUCTEMB and SEMEMB.

| QUERY | STRUCTEMB | SEMEMB |
|---|---|---|
| $21x - 3\sin(x)$ | $-x\log(x) + 21x$ 
 $-3x\operatorname{acosh}(x) + 21x$ 
 $2x - \frac{16\log(x)}{3}$ 
 $-10x\cos(x) + 21x$ 
 $-xe^x + 21x$ | $210x + 210\sin(x)$ 
 $-60x + 20\sin(x)$ 
 $11x + 11\sin(x)$ 
 $21x + 21\cos(x)$ 
 $-70x + 10\sin(x)$ |
| $\sin(x) + \cosh^2(x)$ | $\cosh^2(x) + \cosh(x)$ 
 $\sin(x) + \cos^2(x)$ 
 $\sin(x) + \tan^2(x)$ 
 $\sin^2(x) + \sin(x)$ 
 $\operatorname{acos}^2(x) + \operatorname{acos}(x)$ | $\cosh^2(x) + \cosh(x)$ 
 $\sin(x) + \cos^2(x)$ 
 $\sin^2(x) + \sin(x)$ 
 $\cos(x) + \frac{1}{\cosh^{10}(x)}$ 
 $\log(x)^{20} + \sin(x)$ |
| $x^2 + 5$ | $x^2 + e^e$ 
 $x^2 + \operatorname{atanh}(x)$ 
 $x^2 + \tanh(x)$ 
 $x^2 + \sinh(x)$ 
 $x^2 + \cosh(x)$ | $x^5 + x^2$ 
 $x^2 + 5$ 
 $x^6 + x^2$ 
 $x^4 + x^2$ 
 $\left(x^2 + 1\right)^5$ |
| $\sinh\left(x^2 + 1\right)$ | $\sqrt{x^2 + \log(9)}$ 
 $\log\left(x^2 + \log(9)\right)$ 
 $\log\left(x^2 + \log(8)\right)$ 
 $\log\left(x^2 + ex\right)$ 
 $e^{x^2 + e^4}$ | $\operatorname{atanh}\left(x^2 + 1\right)$ 
 $-\operatorname{atanh}\left(x^2 + 1\right)$ 
 $\operatorname{atanh}\left(x^2 + 5\right)$ 
 $\operatorname{atanh}\left(x^2 + 2\right)$ 
 $\operatorname{atanh}\left(x^2 + 4\right)$ |
| $2x + 1 + e^{-x}$ | $\log(x) + \log(2) + e^{-x}$ 
 $\log(2x) + e^{-x}$ 
 $e^{-x} + e^{-2x}$ 
 $(-x + \log(2x))^5$ 
 $\left(-x + e^{2x}\right)^4$ | $x^2 + 2x + e^{-x}$ 
 $x^2 + 2x + 1 + e^{-x}$ 
 $2x + e^x + e^{-x}$ 
 $x^2 + 3x + e^{-x}$ 
 $3x + e^x + e^{-x}$ |
| $\sin^2(x) + e^{-x}$ | $\log\left(x^2\right) + e^{-x}$ 
 $\log\left(x^5\right) + e^{-x}$ 
 $\log\left(x^3\right) + e^{-x}$ 
 $\log\left(x^4\right) + e^{-x}$ 
 $\log(2x) + e^{-x}$ | $\sin(x) + e^{-10x}$ 
 $x + \sin(x) + e^{-10x}$ 
 $e^{20x} + \sin(x)$ 
 $e^{10x} + \sin(x)$ 
 $x + e^{10x} + \sin(x)$ |
| $x^2\log(x)$ | $x^2\operatorname{acosh}(x)$ 
 $x^3\log(x)$ 
 $x^2\operatorname{atanh}(x)$ 
 $x^2\operatorname{acos}(x)$ 
 $x^2\operatorname{asinh}(x)$ | $x^4\log(x)$ 
 $x^3\log(x)$ 
 $2x^2\log(x)$ 
 $3x^2\log(x)$ 
 $4x^2\log(x)$ |
| $\log\left(x + \sqrt{x^2 + 1}\right)$ | $\frac{1}{x^2 + x + 2\log(2) + 2}$ 
 $\frac{4}{\frac{2x^2}{3} + 1}$ 
 $\frac{5}{5x^2 + x + 10}$ 
 $\frac{1}{x^2 + x + 1 + 2\log(2)}$ 
 $\log\left(x + \frac{1}{\sec(250)}\right)$ | $\log\left(x^{10} + x - 10 + \frac{1}{\csc(10)}\right)$ 
 $\log\left(\frac{1}{\csc(10)} + \frac{1}{x^{10}}\right)$ 
 $\log\left(x - \frac{10}{\csc(10)}\right)$ 
 $\log\left(x - \frac{10}{\csc(20)}\right)$ 
 $\log\left(\frac{x}{2\log(2)} + x + 1\right)$ |

