# OpenReview forum: "Semantic Representations of Mathematical Expressions in a Continuous Vector Space"
_TMLR — Accepted by TMLR_

### Review · Reviewer_WSit · 2023-04-15

**Summary Of Contributions:**

The authors present an approach for creating representations of mathematical expressions. These representations are intended to be semantic - i.e., they capture the expression’s meaning, rather than its surface form (e.g., the same representation should be produced for “tan(x)” and “sin(x)/cos(x)”). The authors achieve this by training encoder-decoder models whose input-output pairs are pairs of semantically equivalent expressions.

The contributions are:
- C1: They create a dataset of equivalent expressions which can be used to train systems under their approach
- C2: They show that encoder-decoder models can succeed on this task of producing expressions equivalent to the input.
- C3: They perform several analyses of the representations (e.g., nearest-neighbor analyses and analogies) showing that the representations learned in their approach capture the meaning of expressions


**Audience:**

Yes

**Claims And Evidence:**

Yes

**Requested Changes:**

- RC1: Relating to W1 above, I would encourage the authors to revise the framing to more clearly acknowledge the existence of prior work with similar aims (i.e., prior work on processing mathematical expressions). For a paper to be valuable, there is no need for it to be the first one to have tackled a problem, so modifying the framing in this way would not weaken the paper. In fact, it would strengthen it: As it currently stands, I could easily imagine readers objecting to this intro, whereas an intro that frames prior work more clearly would not give readers a reason to feel dissatisfied. I think that a straightforward way this modified framing could work would be: “There remains no consensus about how to process or represent mathematical expressions. Some prior works have encoded expressions based on context, but this is suboptimal because [list reasons]. We instead aim to encode expressions based on their meaning. Some prior work (e.g., Lample and Charton) has indirectly achieved this by training systems to perform mathematical tasks, which likely requires some sort of representation of the semantics, but the focus in such work is not representation but rather performance on the task. We instead focus directly on the problem of representation learning. A couple of prior works have also worked in this area - here is a description of how we differ from them: [describe differences from Allamanis et al. and Liu 2022, and any other relevant ones].” There is no need to frame it in this specific way I have listed, but addressing this concern in some way would be important for securing my recommendation.
- RC2: Relating to W2 above, it would be nice (though not necessarily critical to securing my recommendation) to clarify the motivation for producing semantic mathematical representations. In addition to the motivation I listed under W2, another possible motivation that I can imagine is: “There are lots of domains (e.g., language and vision) for which we would like to generate representations that capture semantics. However, for those domains it is hard to evaluate the success of a proposed approach because we don’t have a clear understanding of the correct meaning. In contrast, we do understand the meaning of math very well; so by working with math, we can perform a controlled and principled analysis of the broader question of semantic representations, a question that is relevant to many domains.” In this regard, it might be worth citing the following work, which uses the same basic approach but for language rather than math (i.e., training an encoder-decoder model to produce semantically equivalent expressions): https://arxiv.org/abs/1809.10267
- RC3: Relating to W4, it would be nice to have a description of how the proposed model differs from prior models that are compared to in experiments. This change would not be critical in securing my recommendation, but would strengthen the work.


**Strengths And Weaknesses:**

Strengths:
- S1: The approach is clear and intuitive - it makes a lot of sense as a way to create semantic representations.
- S2: The dataset that they produce will likely be useful to future researchers working in this domain
- S3: The analyses are extensive, interesting, and convincing for the purpose of demonstrating that the learned representations capture semantics. I especially enjoyed Table 4 and Table 5 (the analyses of nearest neighbors and analogies). Another point I especially liked about these analyses is the comparison between ExpEmb-A (a baseline model trained on autoencoding) and ExpEmb-E (their actual model, trained on producing equivalent expressions), since this comparison provides a nice control.
- S4: The paper is clear and well-written.

Weaknesses:
- W1: I believe that the framing is somewhat misleading with respect to prior work and novelty. The intro starts by saying “there exist no major algorithms for processing mathematical content”, which seems unnecessarily dismissive of the several well-received papers that are cited which do process mathematical content, such as Lample and Charton, Allamanis et al., and Liu 2022. Similarly, parts of the introduction seem to suggest that all prior methods for representing mathematical expressions have encoded them based on context alone, when in fact there have been several prior papers which encoded them based on their structure and/or semantics - these include the three papers mentioned earlier in this paragraph, as well as the following 2 papers which perform analyses that are in a similar spirit to the submitted paper’s analyses: https://arxiv.org/abs/1910.06611, https://www.ncbi.nlm.nih.gov/pmc/articles/PMC8491571/. Also, regarding Lample and Charton’s work, even though their focus is not representations, I think that their model’s performance (particularly Section 4.6 “Equivalent Solutions”) gives reason to believe that their system learned reasonably sophisticated semantic representations, which is why I included it in the list of papers that are relevant for learning semantic mathematical representations.
- W2: The motivation for this work is unclear - why do we want semantic representations of mathematical expressions? I can imagine answers to this question (e.g., standard LLMs are bad at math, and we could solve this problem by giving them better representations), but it would be nice to directly motivate this in the paper instead of leaving it to the reader to figure it out.
- W3: All of the evaluations are intrinsic (just analyzing the structure of the learned representations) rather than extrinsic (testing the usefulness of the representations for downstream tasks). It would be nice to also have extrinsic evaluations. However, I do not believe that this weakness is a serious one, as the paper already has a sufficient quantity of analyses. (Also, note that the authors acknowledge this concern).
- W4: There is no description of the strategies used by the relevant prior work (Allamanis et al., and Liu 2022). This paper compares to those papers, which is excellent, but it would be good to also briefly state how the proposed model differs from what they did, to motivate why we would expect the proposed model to differ from these prior models.

---

> ### Author Response · Authors · 2023-05-01
> **Response to Review WSit**
>
> Thank you for your review and comments.
>
> ### RC1/W1
>
> Thank you for mentioning the relevant papers. Our intention was not to
> be dismissive about the existing work but to mention that there are no
> well-established embedding methods for mathematical content like there
> are for NLP (for example, BERT, BART, T5, and so on) and other modes of
> data. We will update the language and include the relevant work in our
> paper.
>
> We will replace the first paragraph of the "Introduction" section with the following:
>
> > Despite there being well-established search technologies for most other modes of data, the processing of
> mathematical content remains an open problem (Larson et al., 2013). Effective search technologies require
> semantically-rich and computationally efficient representations of mathematical data. Most equation embedding methods have focused on establishing a homomorphism between an equation and its surrounding
> mathematical text (Zanibbi et al., 2016; Krstovski & Blei, 2018). While this approach can help find equations
> used in similar contexts, it is less effective in the following situations: (1) there is a large chunk of data in
> which equations occur without any context (consider math textbooks that contain equations with minimal
> explanation), and (2) in scientific literature, an equation may be used in a variety of disciplines with different
> contexts, and encoding equations based on textual context hampers cross-disciplinary retrieval.
>
> We will add the following to the "Introduction" section:
>
> >  In a similar approach, Zhang et al.
> (2017) trained a Seq2Seq model on sentence paraphrase pairs and used the last encoder hidden state as the vector representation of the input sentence.
>
> We will add the following to the "Related Work" section:
> > Russin et al. (2021) and Schlag
> et al. (2019) have studied the representations learned by a Transformer model, trained on a math-reasoning
> dataset. Lample & Charton (2019) showed that their model generated multiple mathematically equivalent
> solutions for a first-order differential equation, showing a degree of semantic learning.
>
> ### RC2/W2
>
> In addition to possible applications in math problem solving, this work
> is also motivated by its possible applications in information retrieval.
> We mention this briefly in the "Related Work" section and will add the
> following to this section.
>
> >  Learning semantically-rich representations of mathematical content would be useful in learning-based applications, like math problem-solving as well as information
> retrieval.
>
> ### W3
>
> We have included an additional evaluation for the distance analysis, in
> response to the comments from reviewer evvu (response to RC1/W1/W3).
> Please feel free to take a look at our comments there. We will also
> revise the paper to include the same.
>
> ### RC3/W4
>
> Thank you for your suggestion. We will include a brief overview of the
> models used by EqNet and EqNet-L.
>
> > For EqNet, Allamanis et al. (2017) used
> the tree representation of a formula with a modified version of the recursive neural network (TreeNN). They
> partitioned their training set into equivalent classes, and they used this information while computing the
> training loss. They also introduced a regularization term in their objective, called subexpression autoencoder.
> EqNet-L (Liu, 2022) extended EqNet by adding a dropout layer and introducing a stacked version of the
> subexpression autoencoder

---

> > ### Comment · Reviewer_WSit · 2023-05-04
> > **Thank you for the response! This addresses my concerns**
> >
> > Dear authors,
> >
> > Thank you for the response! This response addresses the concerns that I listed, so with these changes I would be happy to recommend the paper for acceptance.

---

> > > ### Author Response · Authors · 2023-05-04
> > > **Minor Update regarding RC1/W1**
> > >
> > > Dear reviewer WSit,
> > >
> > > There is a minor update regarding RC1/W1. After further edits, we have moved the following text segment to the "Related Work" section:
> > > > Analogous to our approach, Zhang et al. (2017) trained a Seq2Seq model on sentence paraphrase pairs and used the last encoder hidden state as the vector representation of the input sentence.
> > >
> > > Apologies for the confusion!

---

### Review · Reviewer_evvu · 2023-04-20

**Summary Of Contributions:**

The authors main goal is to obtain representations of mathematical expressions which capture some notion of semantic similarity. To do so, they first represent a mathematical equation as a sequence of tokens using Polish notation, and then propose two models, both using a transformer architecture: ExᴘEᴍʙ-A is trained in an autoencoder fashion, whereas ExᴘEᴍʙ-E is trained using pairs of equivalent mathematical expressions. (More on this later.) The authors then evaluate these models in several ways:

1. **Equivalent Expression Generation:** Given an input expression $x$, they obtain $K$ output expressions ${y_k}_{k=1}^K$ using beam search on the trained transformer, and evaluate how often an expression in this set is mathematically equivalent to $x$, in the sense that $y_k - x$ can be symbolically simplified to $0$.
2. **Embedding Evaluations:** The authors qualitatively inspect a 2-dimensional PCA projection of the encoding of a variety of mathematical expressions and observe some patterns. Given a query, they also inspect the top 5 embeddings which are closest in cosine distance to the query embedding, and note some patterns. Finally, they explore what is captured in the vector space structure of the embeddings using an analogy-like task.
3. **Similar Representations of Mathematically Equivalent Expressions:** They compare against existing baselines which attempt to learn representations of expressions such that equivalent expressions are most similar to one another. Given a test equation, they calculate the proportion of the $k$ nearest neighbors which are mathematically equivalent to the test equation. The authors underperform the baselines in the majority of these evaluations.

**Audience:**

Yes

**Broader Impact Concerns:**

I do not feel there are any ethical implications of the work that would require adding a Broader Impact Statement.

**Claims And Evidence:**

No

**Requested Changes:**

1. The notion of similarity that the authors are attempting to encode is never defined. The authors do explicitly exclude contextual similarity (i.e. expressions used in similar contexts have similar representations) and say they want to consider "semantic similarity" in addition to mathematical equivalence (end of section 2), however exactly what this means is unclear. The lack of clarity on this point becomes more problematic in the qualitative evaluation (section 5.3). For example, in Table 4 it is stated that "ExᴘEᴍʙ-E does a better job at learning semantics and the overall structure of the expressions", and one of the supporting examples is that ExᴘEᴍʙ-E $4x^2 \cos(3x-1)$ matches to expressions which generally have the form $\alpha x^2 \cos(x + \beta)$, whereas ExᴘEᴍʙ-A matches to expressions which are of the form $4x^2 e^{\alpha x+\beta}$. It is unclear to me why the former is preferred to the latter, and to my understanding there is no clear definition of similarity aside from some gut feeling. This could be excused if the paper was proposing to learn some general representations from a large amount of unsupervised pretraining and then leverage those representations for some downstream tasks, in which case the lack of a well-defined notion of similarity could be justified by saying that a rigorous definition is not possible but clearly the representation is useful in tasks for which some notion of similarity is required, however this is not present in this work. I could imagine several notions of similarity in mathematical expressions. For example, edit distance of the Polish notation where different penalties apply for different types of edits seems closest to what the author's mean by "similarity", and this could be formalized. In other settings, one may be more interested in predefined distances on function spaces (eg. $\mathcal L^k$ distance). In practice, the sort of function representation I find myself most often in need of is the ability to describe a set of properties and obtain functions with those properties, and this sort of similarity would seemingly be currently best supported by the sort of contextual embedding eschewed by the authors at the start. This point is a significant weakness of the paper, in my opinion, and the authors should either seek to define what sort of similarity they are after or defend the lack of such a definition by demonstration of the usefulness of their representation on other tasks of interest which can be well-defined.
2. It is somewhat unsurprising that the representations for ExᴘEᴍʙ-A are relatively weak, as (to my understanding) the model is simply an autoencoder and does not receive any information about mathematical equivalence whatsoever. As such, the relatively strong performance of ExᴘEᴍʙ-A on any tasks should be viewed not as a strength, but rather as a useful critique of an evaluation which actually does not require knowledge of mathematical equivalence, and instead can be solved simply by token similarity. In this light, the relative performance of ExᴘEᴍʙ-A compared to ExᴘEᴍʙ-E in table 3 vs. table 6 makes complete sense. As such, I take issue with the statement at the end of page 6 that "both ExᴘEᴍʙ-A and ExᴘEᴍʙ-E are capable of learning the mathematical structure", and would encourage the use of ExᴘEᴍʙ-A not as a baseline representation model but rather as a model which is able to discern which test examples do *not* require mathematical understanding in order to get them correct (eg. as a consequence of unintended dataset / task bias).
3. I would like clarification on how ExᴘEᴍʙ-E is trained. As I understand it, given a set of equivalent expressions $\{x_k}_{k=1}^K$ then during training ExᴘEᴍʙ-E would attempt to model $f(x_i) = x_j$ for every $i \ne j$. One way to interpret this is thinking of ExᴘEᴍʙ-E as an autoregressive language model, where the data looks like strings $x_i = x_j$, and thus for input $x_i$ we want the continuation $x_j$. Is this correct? If so, why not try a contrastive training process, i.e. minimize $d(f(x_i), f(x_j))$ while maximizing $d(f(x_i), f(y))$ for some $y$ which is not equivalent to $x_i$? It is not that your current approach for ExᴘEᴍʙ-E is wrong, per-se, but the properties of natural language vs. mathematical expressions seem significantly different enough that it almost seems like banging a nail in with a drill. A contrastive loss seems much more in-line with your stated objective. Moreover, if you had a crisp definition of similarity, you could even sample $y$ in accordance with this (with probability inversely proportional to the edit distance, for example).
4. As mentioned above, the qualitative analysis needs to be improved, and the analogy task needs to be rethought.

**Strengths And Weaknesses:**

**Strengths**
1. The paper is attempting to solve an important problem - finding a useful embedding of mathematical expressions.
2. Paper is clearly written and easy to understand. The authors do a good job of motivating the problem of mathematical expression representation and why it differs from language modeling.
3. The authors include a variety of ablations and compare against reasonable baselines.
4. The authors include a sincere discussion of limitations in the conclusion.

**Weaknesses**
1. The notion of "similarity" the authors are after is unclear. This is the most serious issue with the current work, and I go into more detail on this in the "Requested Changes".
2. The models chosen seen suboptimal for the author's purpose. At a high level, ExᴘEᴍʙ-A is trained as an autoencoder on mathematical expressions without any signal for equivalent mathematical structure, and ExᴘEᴍʙ-E essentially treats the problem as autoregressive language modeling on strings which start with the Polish notation form of an expression followed by "=", but neither of these seem ideal for learning representations of mathematical expressions with some level of semantic similarity. (More on this in "Requested Changes".)
3. The qualitative analysis of the embeddings is a bit weak. For example, the authors state "ExᴘEᴍʙ-E groups expressions with operators of the same type together, indicating its ability to understand semantics, whereas ExpEmb-A groups expressions mainly based on their visual structure", but no crisp definition of visual structure was given. Does this mean, for example, that ExᴘEᴍʙ-A considers $(x+1)^2$ to be more similar to $(x+5)^2$ than, say, $x^2 + 2x + 1$? Formalizing this would also present more quantitative analyses.
4. The "analogy" task also fits in the above point, but deserves it's own discussion. I think the authors would be hard-pressed to defend this interpretation of the vector space structure. It seems as though the analogy has two interpretations: (1) $y_1$ is equivalent to $x_1$, with some particular style of equivalence semantically clear to the authors which they would like to apply to $y_2$ to obtain $x_2$, or (2) $y_1 = f(x_1)$, and the authors would like $y_2 = f(x_2)$. Consider what the vector $v = x_1 - y_1$ means in both cases. The first interpretation is hard to justify - recall that semantically equivalent expressions are already intended to be nearby each other. What happens if we apply this vector more than once? For example, if $v$ is the vector for $x_1 = \sin(-x)$ and $y_1 = -\sin(x)$, then if we add this to $\cos(x)$ we expect to end up at $\cos(-x)$. What happens if we add this to $\cos(-x)$? It seems reasonable to expect that we end up at $\cos(x)$ again, but clearly we can't. The second case has a similar problem: consider $v$ to be the vector for $x_1 = x^2$ and $y_1 = x$, then the vector $v$ would represent the sqrt operation (as indicated by the fact that the desired $x_2 = \sin^2(x)$ for $y_2 = \sin(x)$). If we assume such a $v$ exists, we can easily see it must be $0$, as applying it to $y_2 = 1$ would need to result in $x_2 = 1$, but then this is problematic since it would imply that the square of any expression has the same representation as the expression itself. There are other issues apart from this (eg. what happens when the same operation is achievable by two different vectors, such as a $v'$ representing division by $x$, in which case $v = v'$ if we consider $x_1 = x^2$ and $y_1 = x$, but also $v = 2v'$ if we consider $x_1 = x^4$ and $y_1 = x^2$) which makes the analogy setup quite problematic.
5. The Equivalent Expressions Dataset seems to be a rather weak evaluation of the desired goal. It seems like the expressions are fairly clearly separated into a few classes (based on Table 1) making it easy to distinguish many non-equivalent expressions purely based on the tokens they contain. Additional statistics (eg. number of equivalence classes, histogram based on size of equivalence class) would be useful in getting a high-level view of the difficulty. Some examples of equivalence classes added in the appendix would also be appreciated. Personally, I feel like the dataset itself misses the mark when it comes to semantically equivalent mathematical expressions. ML models which operate on statistical co-occurrance patterns are ill-suited to encoding equivalence which can be obtained symbolically, as is the case with the samples in this dataset. More often than not the difficulty is not that equations are written unreduced or expanded, in which case they can be simplified to the same form. In most cases there is a canonically or conventionally "simplest form", but different authors may use different names for variables, functions, and notation, which seems better suited to statistical co-occurrance models but which this dataset does not address.
6. The model performance on the only task which arguably has a well-defined objective (identifying equivalent mathematical expressions, section 5.4) is relatively poor. It outperforms the baseline on 6/14 datasets, 2 of which only marginally so. There is a correlation with training set size with the exception of the BᴏᴏʟL5 dataset, which is particularly notable since it so drastically underperforms whereas it achieves perfect results on Bᴏᴏʟ8. In my opinion, it would be worth trying to understand why it achieves perfect results on Bᴏᴏʟ8 beyond scale of the training set - i.e. did the model actually learn to generalize appropriately, or is it just a very easy dataset/task for some reason?

---

> ### Author Response · Authors · 2023-05-01
> **Response to Review evvu (1/2)**
>
> Thank you for your review and comments.
>
> ### RC1/W1/W3
>
> In this work, we define semantic similarity based on the operators
> present in the expressions. For example, we consider $\sin(x + 1)$ to be
> more similar to $\sin(x)$ than $\log(x)$ or $\sinh(x + 1)$. We do agree
> that a quantitative measure for similarity would be useful. Inspired by
> your comment, we used Tree Edit Distance [1] between the
> operator trees of two expressions as a measure of similarity. For
> this experiment, we considered 2,000 expressions and searched for the
> most similar expression from a pool of expressions, based on the
> similarity between their representations. We computed Tree Edit Distance
> between the query expressions and results and observed the following:
>
> * *Scenario 1.* If we compare the expressions in their original form,
> ExpEmb-A returns an expression strictly closer to the
> query in 1,044 cases and ExpEmb-E in 261 cases.
>
> * *Scenario 2.* When we exclude the constant multipliers and additions, i.e., we
> consider $a f(x) + b$ and $f(x)$ to be at a distance of zero,
> ExpEmb-E returns an expression strictly closer to the
> query in 683 cases and ExpEmb-A in 402 cases. Note that
> $a$ and $b$ could be simple constants, like $1$, $\sqrt{2}$, etc or
> constant sub-expressions, like $\cos(1)$, $\cos(\log(1))$, etc.
>
> For example, consider three expressions $A = \tan(x + \cos(1) + 1)$,
> $B = \tan(x - 40)$, and $C = \exp(x + \cos(1) + 1)$. If we compare them
> in their original form, $A$ and $C$ are closer. But if we consider
> $x + \cos(1) + 1$ and $x - 40$ to be at a distance of zero, $A$ and $C$
> become closer.
>
> It is interesting to see the change in results as we move from the first
> scenario to the second one. These results indicate that
> ExpEmb-E puts more emphasis on operators compared to
> ExpEmb-A. The five closest expressions to the query
> returned by ExpEmb-E and ExpEmb-A show a
> similar trend. Depending on the application, either of these scenarios
> could be considered favorable. In this work, we care more about the
> operators, as they relate to the notion of semantics.
>
> Again, we would like to thank the reviewer for this comment. We believe
> this analysis is very useful when analyzing the representations learned
> by different models and will definitely include these results in the
> revised version of the paper.
>
> ### RC2
>
> We agree with your statement and our intention of adding
> ExpEmb-A in the experiments was to show a contrast between
> the representations learned by ExpEmb-E and
> ExpEmb-A. When we say that ExpEmb-A is
> capable of learning mathematical structure, we mean that a Seq2Seq model
> can learn to generate an expression exactly as the input and has the
> capability of generating mathematical expressions. We will amend the
> statement that you are referring to:
>
> > This experiment demonstrates that both ExpEmb-A and ExpEmb-E are capable
> of generating structurally correct mathematical expressions, and
> ExpEmb-E can generate visually different but mathematically equivalent
> expressions.
>
>
> ### RC3/W2
>
> Yes, your understanding of the training process is correct. We used this
> training process and not contrastive learning because this type of
> training process has been shown to learn generic representations in NLP.
> Also, the intention was to show that ExpEmb-E could learn
> semantically rich expressions without explicitly pushing representations
> of non-equivalent expressions apart during training. Furthermore, this
> method of training could be extended to the case where surrounding text
> is also included in the training process (for example, a corpus of
> mathematical text where more training examples are generated by
> replacing mathematical expressions with their equivalents). Having said
> that, a comparison with contrastive learning would be interesting, and
> this comparison could be quantified using Tree Edit Distance (in
> addition to the SemVec datasets), thanks to your suggestion. We will
> explore this in our future work.
>
>
>
> [1] Kaizhong Zhang and Dennis Shasha. Simple fast algorithms for the editing distance between trees and related problems. SIAM Journal of Computing, 18:1245–1262, 1989.

---

> > ### Author Response · Authors · 2023-05-01
> > **Response to Review evvu (2/2)**
> >
> > ### RC4/W3
> >
> > The analogy task is defined similarly to analogies in NLP. We agree that
> > there are known issues with analogies, for example,
> > <https://aclanthology.org/S17-1017.pdf>, and they alone should not be
> > considered as the proof of efficacy. For the purpose of this
> > paper, the analogies should be interpreted as $x_1$ is similar to $y_1$
> > in the same way $x_2$ is similar to $y_2$. For a given $x_1$, $y_1$, and
> > $y_2$, we find an expression whose embedding is *closest* to
> > $z = \textrm{emb}(x_1) - \textrm{emb}(y_1) + \textrm{emb}(y_2)$. As the
> > embedding vector of $x_2$ is not the same as $z$ and only closest to $z$
> > compared to other expressions in the pool, we do not think we can assign
> > a meaning to applying the difference $x_1 - y_1$ twice to $y_2$.
> >
> > Further, these results may not have a direct application for any
> > downstream task, and the analogies do not always work, as shown in Table
> > 5 in the paper. But this task shows an interesting property of the
> > learned representations, and we feel it is worth mentioning in the
> > paper.
> >
> > ### W5
> >
> > The Equivalent Expressions Dataset is not divided into equivalent
> > classes. Each example in the dataset consists of a pair of equivalent
> > expressions. Table 1 in the paper shows the number of expressions
> > containing a particular class of operators, but the expressions may
> > contain operators from multiple operator classes. For example, 1,424,243
> > expressions consist of operators from a single class, 1,186,808
> > expressions contain operators from two classes, and 130,547 expressions
> > consist of operators from three classes, and so on. The below table
> > shows some examples from the dataset.
> >
> >
> > | Input | Output |
> >  -------|----------
> >  | $1 \cdot \frac{1}{2} x^{\frac{5}{2}} - \frac{5 x^{\frac{3}{2}}}{2}$ | $1 \cdot \frac{1}{2} x^{\frac{3}{2}} (x - 5)$ |
> >  | $1 \cdot \frac{1}{2} x \log{(x )} + 2 x + \frac{5}{2}$ | $1 \cdot \frac{1}{2} (x \log{(x )} + 4 x + 5)$
> > |  $x + e^{x} - 20 + \frac{1}{\sec{(x - \frac{\pi}{2} )}}$ | $x + e^{x} + \sin{(x )} - 20$ |
> > |  $x^{2} (\cos^{-1}{(x )} + 1) + x + 3$ | $x (x \cos^{-1}{(x )} + x) + x + 3$ |
> > | $\frac{5 \pi x^{4}}{4} + 1 \cdot \frac{1}{4} \pi x$ | $1 \cdot \frac{1}{4} \pi (5 x^{4} + x)$
> > |  $x + \log{(x^{2} )} - 1 + 2 \log{(5 )}$ | $x + \log{(25 x^{2} )} - 1$ |
> > |  $5 e^{x} + \log{(x^{\frac{5}{2}} )}$ | $\frac{5 \cdot (2 e^{x} + \log{(x )})}{2}$ |
> > |  $- 3 x \cos^{-1}{(x )} + 4 x$ |  $3 x (1 - \cos^{-1}{(x )}) + x$ |
> > |  $2 (\tanh^{-1}{(x )} + 1) \log{(2 )}$ | $(\tanh^{-1}{(x )} + 1) \log{(4 )}$ |
> > |  $- \frac{10 x}{\coth{(\sin{(10 )} )}} + 10$ | $- 10 (x \tanh{(\sin{(10 )} )} - 1)$ |
> > |  $- 20 \sin{(x )} + 20 \cosh{(x )}$ | $- 20 \cos{(x - \frac{\pi}{2} )} + 20 \cosh{(x )}$ |
> > |  $(- x + \sinh^{-1}{(3 )}) (x + 1)$ | $- (x + 1) (x - \sinh^{-1}{(3 )})$ |
> > |  $x \cos{(x )} + \cosh^{-1}{(x )}$  |  $x \sin{(x + 1 \cdot \frac{1}{2} \pi )} + \cosh^{-1}{(x )}$ |
> > |  $1 \cdot \frac{1}{4} x + 1 \cdot \frac{1}{4} + 1 \cdot \frac{1}{4} \cdot \frac{1}{x} \log{(x )}$ |  $1 \cdot \frac{1}{4} \cdot \frac{1}{x} (x^{2} + x + \log{(x )})$ |
> > |  $x \tan^{-1}{(3 )} + 1 \cdot \frac{1}{2} \tan^{-1}{(3 )}$      |     $(x + 1 \cdot \frac{1}{2}) \tan^{-1}{(3 )}$ |
> >
> >
> > We will include these examples in the Appendix of the paper.
> >
> > ### W6
> >
> > We believe the training set size is the main reason for the low
> > performance on some datasets as the Transformer model has a higher
> > number of parameters compared to the existing methods. In our early
> > experiments, we used a simple GRU-based seq2seq model (not included in
> > the paper), and the performance on the datasets with smaller training
> > set sizes (SimpBoolL5, oneV-Poly10, and
> > Bool5) was better compared to the Transformer
> > architecture. Even for BoolL5, the performance was better
> > with a simple GRU-based model. So the model performance seems to be
> > correlated with the training set size, and the datasets with a smaller
> > training size work better with simple architectures. We chose to not
> > vary the architectures and hyperparameters across datasets for a fair
> > comparison with the existing methods.
> >
> > As you mentioned, there are other factors at play as well. For example,
> > BoolL5 dataset seems to be the hardest dataset with 10
> > variables, 7,312 equivalent classes, and 36,050 expressions in total
> > compared to Bool8 which has 3 variables, 120 equivalent
> > classes, and 39,048 expressions. Out of all datasets,
> > Bool10 has the same number of variables as
> > BoolL5, but it has significantly higher expressions per
> > equivalent class, resulting in a larger training set size for
> > ExpEmb-E and hence better performance.

---

> > ### Comment · Reviewer_evvu · 2023-05-23
> > **Defining similarity is crucial**
> >
> > First, let me sincerely apologize for the lateness of my reply.
> >
> > Thank you for your detailed rebuttal and the additional analyses you have conducted to address the concerns raised.
> >
> > Based on the [acceptance criteria for TMLR](https://jmlr.org/tmlr/acceptance-criteria.html), your submission must provide clear evidence supporting your claims, which should also be of interest to the audience of this journal. In this regard, the two primary claims your paper makes are:
> >
> > 1. A Seq2Seq model can learn to generate expressions mathematically equivalent to the input.
> > 2. The encoder of this model captures semantics and excels at clustering and retrieving similar mathematical expressions.
> > In addition, your submission also introduces a corpus of equivalent algebraic and transcendental expression pairs for future research.
> >
> > After evaluating your responses, I am convinced that you have provided sufficient "proof-of-concept" level support for the first claim. I find the justification for the second claim to be less compelling, however, primarily due to the lack of a concrete definition of similarity, as mentioned in my original review.
> >
> > Despite the additional tree edit distance analysis you have provided, the findings from scenario 1 somewhat contradict your second claim. The results from scenario 2 offer some support for this claim in a context where expressions are considered equivalent up to affine transformation. If your paper had a motivating reason for considering this notion of similarity at the fore this would be strong supporting evidence, however since the work did not intend to discover such a similarity it seems more like confirmation bias than objective evaluation.
> >
> > To make a compelling case for the second claim, a well-defined and rigorous notion of similarity should be established from the outset. This notion should ideally be motivated by a specific use-case, as different situations will necessitate different concepts of similarity between mathematical expressions. As it currently stands, the evidence supporting the second claim still rests on loosely defined notions of similarity.
> >
> > Thus, while I appreciate your efforts to respond to the initial review, I maintain that the second claim requires a clearer definition of similarity for its validation. As such, at this time I am "leaning reject", however I would encourage you to resubmit once you have identified a specific notion of similarity between mathematical expressions which is motivated by a particular use-case for which the contextual similarity provided by LLMs falls short.

---

> > > ### Author Response · Authors · 2023-05-23
> > > **Response to the Comment from Reviewer evvu**
> > >
> > > Dear reviewer evvu,
> > >
> > > Thank you for your comment.
> > >
> > > As the notion of similarity considered in the paper was not clear in our initial submission, we revised the paper to explicitly include that the paper considers similarity based on the operators present in the expressions (mentioned at the end of Section 2).
> > > > We define semantic similarity based on the operators present in expressions. For example, we consider $\sin(x + 1)$ to be more similar to $\sin(x)$ than $\log(x)$ or $\sinh(x + 1)$.
> > >
> > > With this definition, we believe the results presented in the "Distance Analysis" section provide evidence for the second claim. Scenarios 1 and 2 indicate that ExpEmb-E puts more emphasis on the operators while representing an expression as a vector compared to ExpEmb-A. This additional evaluation corroborates our initial results presented in Table 4. We agree that tree edit distance is not a perfect measure, and we have mentioned this as one of the limitations of the paper.
> > >
> > > Furthermore, as mentioned at the end of Section 2, a motivation behind the proposed embedding scheme is information retrieval where operator-based similarity may prove useful (in addition to being able to retrieve mathematically equivalent expressions).
> > > > Our approach considers semantic similarity in addition to mathematical equivalence. Learning semantically-rich representations of mathematical content would be useful in learning-based applications, like math problem-solving as well as information
> > > retrieval.
> > >
> > > As we want to compare the overall structure of the mathematical expressions in terms of operators, tree edit distance is a reasonable metric for comparing the similarity. Thus, we disagree with the statement that these results are merely a confirmation bias.

---

> > > > ### Comment · Reviewer_evvu · 2023-05-26
> > > > **Response re: ambiguous definition of similarity**
> > > >
> > > > Thanks for the additional clarification, but your definition of similarity is still not rigorously defined.
> > > >
> > > > You mention that you define semantic similarity based on the operators present, and go on to say that $\sin(x+1)$ is more similar to $\sin(x)$ than $\log(x)$ or $\sinh(x+1)$, which is a fine example, but is $\sin(x+1)$ more similar to $\sin(x)$ than $\sin(x+5)$ or $\sin(2x)$? Perhaps these are all considered "equivalent" in your setting, but the motivation for considering them equivalent is unclear to me. If they, indeed, are equivalent, then $\cos(x)$ should also be considered equivalent to $\sin(x)$, since it is off by just a phase shift, however based on the examples I don't believe you consider these to be equivalent. Then again, if it is not equivalent surely it is more similar to $\sin(x)$ than $\log(x)$.
> > > >
> > > > Your motivation at the end of Section 2 is good, but your proposed definition of similarity does not, in my opinion, address that definition.

---

> > > > > ### Author Response · Authors · 2023-05-26
> > > > > **Response to the Comment from Reviewer evvu**
> > > > >
> > > > > Dear reviewer evvu,
> > > > >
> > > > > Thank you for your comment.
> > > > >
> > > > > We consider $\sin(x + 1)$, $\sin(x)$, $\sin(2x)$, and $\sin(x + 5)$ to be at a distance of 0 for the distance analysis. Ideally, we would like to consider them at different distances, but defining the penalty for calculating the tree edit distance becomes complicated, for example, should $\sin(x)$ be considered closer to $\sin(x + \log(1))$ or $\cos(x)$? Furthermore, if a user searches for $\sin(x)$, they are more likely to assign a higher score to a retrieval system that returns the expressions of form $\sin(ax + b)$ than a system that returns $\cos(x)$, where $a$ and $b$ are constant expressions. Hence, we consider $\sin(ax + b)$ to be at a 0 tree edit distance from $\sin(x)$ for simplicity. And we consider $\sin(x)$ and $\cos(x)$ to be at a tree edit distance of 1.
> > > > >
> > > > > Regarding $\cos(x)$ and $\log(x)$, yes, $\cos(x)$ can be considered more similar to $\sin(x)$. But that would require defining different penalties for different tokens, where a token can be an operator, a variable, or a constant. We feel it would add even more bias to an already imperfect evaluation metric. So we decided to keep a penalty of 1 for all additions, deletions, and updates, with the exception that $f(ax + b)$ is considered to be at a distance of 0 from $f(x)$.

---

> > > > > > ### Comment · Reviewer_evvu · 2023-05-26
> > > > > >
> > > > > > I think your response highlights the issue. To be clear, my criticism is not that the tree-edit distance metric is ill defined, but rather that the connection between the similarity required for the motivating setting at the end of section 2 and that which the model is shown to exhibit is tenuous at best. Whether a user would be happier with retrieving expressions of the form $\sin(ax+b)$ or $\cos(x)$ when issuing a query of $\sin(x)$ depends entirely on a clearer understanding of the setting in which this query is being performed, and thus a clearer distinction of the sort of similarity which is required, at which point an appropriate metric would suggest itself. In particular, my interpretation of your motivating statement did not suggest that a reasonable definition of similarity would be expressions which simply use the same operators as the query expression.
> > > > > >
> > > > > > In my reading of your paper, I see this claim being supported in the reverse direction - two embedding methods were proposed, some examples of similar expressions from each were inspected, and the authors subjective notion of similarity was better captured in the second than the first. As I highlighted in my original review, one could easily provide reasons why expressions which were considered similar by the first model were more similar to each other than the second. Without a clear definition of what it means to be similar in the context of your motivating statement it is not possible to provide sufficient evidence for a claim that the model does a better job of capturing similarity.

---

> > > > > > > ### Author Response · Authors · 2023-05-27
> > > > > > > **Response to the Comment from Reviewer evvu**
> > > > > > >
> > > > > > > Dear reviewer evvu,
> > > > > > >
> > > > > > > Thank you for your comment.
> > > > > > >
> > > > > > > We agree that the definition of similarity may vary from application to application, and one definition may not be universally applicable, which is why we clarify the definition and limitations in the paper. We believe the results presented in the paper support the claims, given this definition. Furthermore, the distance analysis is one evaluation among other evaluations, like embedding plots, analogy tasks, and a comparison on with the existing methods on the SemVec datasets.

---

> > > > > > > > ### Comment · Reviewer_evvu · 2023-05-31
> > > > > > > >
> > > > > > > > As mentioned above, I disagree that a clear definition has been provided. Initially, no definition was provided at all. After the initial review, a general notion ("We define semantic similarity based on the operators present in expressions.") was provided, but this is not a definition. An example is provided ("$\sin(x+1)$ is more similar to $\sin(x)$ than $\log(x)$ or $\sinh(x+1)$") but as discussed above there are a number of ambiguities with this definition. Moreover, I do not feel this notion is in alignment with the stated motivation to learn "semantic similarity" at the end of section 2. This strikes me as analogous to claiming to capture semantic similarity between words by considering words similar if they contain the same letters.
> > > > > > > >
> > > > > > > > Apart from the inherent issues in the definition, there is also a scientific issue in the way the definition was proposed. I think you would agree that proposing a definition after having already made a claim about which model better preserves similarity makes the potential for confirmation bias quite likely. If the complications of such a setting was not enough to give one pause, the fact that ExᴘEᴍʙ-A performed better on the tree-edit distance analysis prior to ignoring differences from linear transformations (which heretofore did not come up as an explicit objective) should have really made this a poignant concern.
> > > > > > > >
> > > > > > > > To be clear, my issue is not with the distance analysis itself, but rather with the fundamental disconnect between the claim of capturing semantic similarity and the ambiguous notion of similarity which was proposed after-the-fact. In my opinion, this makes the claim that ExpEmb-E better captures semantic similarity insufficiently supported.

---

### Review · Reviewer_xb2r · 2023-04-23

**Summary Of Contributions:**

The paper presents a new approach for representing mathematical expressions in a continuous vector space, such that distances in the embedding space reflect more about semantic similarity in math than similarity in the surface form.

The paper proposes to obtain such representations by training an enc-doc Transformer to generate mathematically equivalent expressions, and demonstrates that the resulting encoder hidden states (EXPEMB-E) turn out to be reasonable representations for downstream retrieval and analogy tasks. The training task of generating mathematically equivalent expressions is shown to be more effective than an autoencoding task (EXPEMB-A)  because the resulting representations are better structured according to operator types.

**Audience:**

Yes

**Broader Impact Concerns:**

No particular concern on ethical implications.

**Claims And Evidence:**

Yes

**Requested Changes:**

The paper mentions that encoding expressions using surrounding textual information could potentially hinder downstream retrieval performance. But there isn't a comparison with such a text-conditioned baseline.

It would also be great to investigate if it's easier to adapt a pre-trained Transformer into encoding expressions by injecting certain abilities through fine-tuning.

The proposed adjustments won't affect my overall rating for acceptance, but will greatly strengthen the work in my view.

**Strengths And Weaknesses:**

Strengths
- The paper proposes an approach, EXPEMB-E, to represent mathematical expressions *independent from* their surrounding text to support cross-disciplinary retrieval purposes.
- The paper demonstrates the advantage of the proposed EXPEMB-E approach via a suite of qualitative evaluations including 1) the emergence of clusters correponding to different operator types; 2) the alignment between embedding distance and semantic distance for retrieval tasks; 3) the possibility to perform arithmetics in the representation space and retain semantic meaningfulness. E.g. x1 is to y1 as x2 is to y2 --> derive x2 as emb(x1) - emb(y1) + emb(y2).
- The paper quantitatively benchmarks the performance of EXPEMB-E on a previous equivalent expression dataset (Allamanis et al. 2017) and shows that EXPEMB-E outperforms both the baseline EXPEMB-A and previous math-expression representation methods.
- The paper releases a corpus of equivalent transcendental and algebraic expression pairs.

Weaknesses
- The released training corpus only contains univariate algebraic and transcendental operators. Future work is expected to extend it to multi-variate expressions with more complex operators.
- There lacks a comparison between encoding standalone expressions vs. encoding expressions conditioned on the surrounding text.
- The Transformer enc-dec was trained from scratch whereas there could be a potential to utilize mathematical capacity already present in large-scale pre-trained Transformers.

---

> ### Author Response · Authors · 2023-05-01
> **Response to Review xb2r**
>
> Thank you for your review and comments.
>
> ### W1
>
> We agree that the corpus contains simple expressions currently. The
> intent of this work was to investigate whether the proposed approach
> indeed learns semantically-rich representations. As we mention in the
> future work section, revising the corpus to include multivariate
> expressions and more complex operators will be a part of future work.
>
> ### W2/RC1
>
> A comparison between representations learned with and without the
> surrounding text would be a nice addition to the paper. To achieve this,
> the corpus would need to include multivariate expressions and complex
> operators, similar to textbooks and manuscripts. We will consider this
> part of the future work with a new version of the corpus.
>
> ### W3/RC2
>
> We agree that using pre-trained weights rather than random
> initialization could prove useful in achieving better results on the
> tasks presented in the paper. The reason for training the models from
> scratch was to avoid any bias that could be injected into the learned
> representations, as the primary motivation of this work was to
> investigate learned representations.

---

### Author Response · Authors · 2023-05-04
**Revised Paper**

We thank the reviewers for their comments. They were extremely useful and helped us improve the paper. We have replied to the reviewer's comments individually and have revised the paper to address these comments. Below is the summary of major changes in the paper:

1. The first and second paragraphs of the "Introduction" section are updated in response to the comments from reviewer WSit (Page 1).
2. We have added relevant papers mentioned by reviewer WSit to the "Related Work" section (Page 3).
3. We have included an additional evaluation using Tree Edit Distance to address the comments of reviewer evvu (Page 8).
4. The conclusion is updated to reflect the limitations of using Tree Edit Distance (Page 11).
5. A few examples from the Equivalent Expressions Dataset are added to Appendix B in response to comments from reviewer evvu (Page 14).
6. We have moved Figure 2 from Page 6 to Page 7, Table 4 from Page 7 to Page 8, and Table 5 from Page 8 to Page 9 to align them with the corresponding text.

---

### Decision · Action_Editors · 2023-06-11

**Recommendation:** Accept with minor revision

**Comment:**

I believe the manuscript can be repositioned (see claims & evidence).

**Audience:**

There is a large community interested in LLMs for code and mathematical understanding that will be interested in this work, but this means I also recommend expanding the code generation related work.

**Claims And Evidence:**

The work makes three claimed contributions:
1. A corpus of equivalent mathematical expressions
2. Two models (s2s) evaluated on their ability to produce mathematical equivalent expressions
3. Evidence that the continuous vector representations capture semantics and not simply visual structure

I think there is a valid contribution in this work, but I think this is narrower than the claims stated.  There are a number of suggestions in the reviews but I will focus on a few points here.

(1) Is definitely satisfied though I would appreciate more details on the creation process as the distributions (scales) are very different between types and it's not clear to me how expression lengths and related details were decided.

(2) These models are introduced, and evaluated.  The results are interesting but again, I'll not several details I'd like included:
  - What is the effect (if any) of using a pretrained model for initialization
  - What strategies (e.g. for diversity) were employed in the beam search for generation? and how do these effect performance?

(3) I think this is the claim w/ the weakest evidence and I think it should be removed or changed to an analysis in the final version.

As a final note, I would point out that the definition of similarity used should be brought to the forefront of the discussion (and expanded) as it is key to whether future researchers will choose to extend this work or position themselves differently relative to it.

---

> ### Author Response · Authors · 2023-09-01
> **The camera-ready version**
>
> Dear Action Editor,
>
> We have submitted the camera-ready version of our paper. Please let us know if anything else is required from our side.
>
> We would like to thank the anonymous reviewers and you for insightful discussions and feedback, which helped us improve this work.
>
> Thank you,\
> Authors